# Connecting Pre-trained Language Models and Downstream Tasks via Properties of Representations

**Chenwei Wu**
Duke University
cwwu@cs.duke.edu

**Holden Lee**
Johns Hopkins University
hlee283@jhu.edu

**Rong Ge**
Duke University
rongge@cs.duke.edu

## Abstract

Recently, researchers have found that representations learned by large-scale pre-trained language models are useful in various downstream tasks. However, there is little theoretical understanding of how pre-training performance is related to downstream task performance. In this paper, we analyze how this performance transfer depends on the properties of the downstream task and the structure of the representations. We consider a log-linear model where a word can be predicted from its context through a network having softmax as its last layer. We show that even if the downstream task is highly structured and depends on a simple function of the hidden representation, there are still cases when a low pre-training loss cannot guarantee good performance on the downstream task. On the other hand, we propose and empirically validate the existence of an "anchor vector" in the representation space, and show that this assumption, together with properties of the downstream task, guarantees performance transfer.

## 1 Introduction

Large-scale pre-trained language models have achieved strong performance in a wide range of downstream tasks, including natural language inference [Devlin et al., 2018] and reading comprehension [Brown et al., 2020]. For many of these tasks, training a linear classifier on top of the hidden-layer representations generated by the pre-trained models can already provide near state-of-the-art results [Belinkov et al., 2017]. Despite many empirical investigations about the zero-shot applications of these pre-trained models, there is little theoretical understanding of their empirical success. In this paper, we aim to theoretically investigate this core question:

> *When can the **representations** from pre-trained models transfer to downstream tasks that are **very different** from the pre-training task?*

This is a fundamental question in understanding why good performance in pre-training leads to good performance on downstream tasks. Unlike the notion of generalization in traditional learning theory where the models are evaluated in the same task and the test data are sampled from the same distribution as the training data, here the downstream tasks are usually very different from pre-training. For instance, people can pre-train large language models using cross-entropy loss on a language modeling task with webpage data, and evaluate the models using classification accuracy on text classification in news articles. The differences between pre-training and downstream tasks make it challenging to explain the success of these language models.

37th Conference on Neural Information Processing Systems (NeurIPS 2023).

To overcome this challenge, we need a way to model the relationship between the pre-training and downstream tasks. Previous research has taken several approaches in this direction: Wei et al. [2021] assumes a latent-variable generative model for the data and a downstream task depending on the latent variables; Saunshi et al. [2021] formulates the downstream classification task as a language modeling task which is similar to the pre-training task. These works either rely on strong explicit assumptions about the structure of the data (i.e., assuming the data is generated from a simple generative model) or treat the entire pre-trained model as a black box.

## 1.1 Our contributions

In this paper, we consider a very general model for the data and open the black box of the pre-trained model at the last layer. Specifically, for an input sequence $x = (x_1, \ldots, x_L)$ where the entries comes from a dictionary $\{1, \ldots, n\}$, we assume the observation probability of $x_i$ satisfies a log-linear model

$$p^*(x_i = j | x_{-i}) \propto \exp(\langle v^*_{-i}(x_{-i}), v^*_j \rangle),$$

where $x_{-i}$ is the sequence $x$ without $x_i$, $v^*_j$ is a vector only depending on word $j$, and $v^*_{-i}$ can be an arbitrary function. This aligns with commonly used networks whose last layer is usually a softmax layer. Moreover, since our model does not put any constraint on the function $v^*_{-i}$, it can be arbitrarily complicated, such as a huge transformer model such as BERT [Devlin et al., 2018] or GPT-3 [Brown et al., 2020]. We also allow the distribution of the input to be different in pre-training and downstream tasks. This makes our setting more general than previous latent models Wei et al. [2021], Arora et al. [2016].

We assume the pre-training task is to predict a word from its context. During pre-training, for every input sequence $x$, we want our model to predict the "label" $x_i$ from $x_{-i}$. The model we use in training (which we call the "student model") has the same log-linear structure: $p(x_i = j | x_{-i}) \propto \exp(\langle v_{-i}(x_{-i}), v_j \rangle)$.

For the downstream task, for simplicity we focus on binary sequence classification, e.g., sentiment classification. To define downstream tasks, let the "logits" of the ground-truth model be defined as $z^* := (\langle v^*_{-i}(x_{-i}), v^*_j \rangle)_{j=1}^n$ (these are just the outputs before the softmax computation), and assume that the downstream task is specified by a function of the logits, $f^*(z^*)$.

In reality, we do not have access to the ground-truth model $v^*_{-i}$ and $v^*_j$. Instead, we only have access to the student model $v_{-i}$ and $v_j$ that achieves low pre-training loss. We can define the student logits as $z := (\langle v_{-i}(x_{-i}), v_j \rangle)_{j=1}^n$. In some sense, $z$ is the representation learned by the pre-training step. A natural idea for solving the downstream task would be to learn a function $f(z)$. More details about this model will be provided in Section 2.

Our goal is to understand the properties of the learned representation. In order to simplify the problem, we assume that the student model can be optimized to achieve a small KL divergence with the true word probabilities during training. Under this setting, the question we ask above becomes:

> *If the downstream task depends on a simple function of the logits $f^*(z^*)$ and we have access to a student model $p$ such that $\mathbb{E}_x[D_{\mathrm{KL}}(p^*(x_i|x_{-i})||p(x_i|x_{-i}))]$ is small, under what conditions is there a function $f$ such that $f(z) \approx f^*(z^*)$?*

A priori, one might imagine if the function $f^*$ is very simple (e.g., a linear function), then it should be easy to find a function $f$. However, we give two counter-examples that show there are additional properties that $f^*$ needs to satisfy: (i) $f^*$ should not distinguish between words with small probabilities and words with super-small probabilities, and (ii) the hidden representations must have some structure that deals with the shift-invariance of the softmax function (that is, the result of softmax does not change if all the logits are shifted by the same constant). We present the counterexamples in Section 3.

To further investigate the structure of the hidden representations and see how we can deal with the shift-invariance property of softmax, in Section 4, we propose and empirically verified the "anchor vector hypothesis": there exists an "anchor vector" in the representation space that can be used to estimate the *bulk partition function*, which we define to be the sum of the exponential of all logits except the largest few, i.e., $\sum_{j: z^*_j \text{ not large}} e^{z^*_j}$. We show how the anchor vector can be used to address the shift-invariance of softmax.

Based on the observation that anchor vectors exist, in Section 5, we give sufficient conditions that shows when a sparse one-hidden-layer ReLU network $f^*$ can be learned by our student model $f$. Specifically, assuming that $f^*$ is a one-hidden-layer ReLU network depending on a small set of words and the downstream task is binary classification depending on $f^*(z^*)$, the existence of the anchor vector enables us to upper bound the loss on the downstream task attained by the student model $f(z)$, in terms of its KL divergence in pre-training. In other words, a small pre-training loss is guaranteed to transfer to a small downstream classification error.

## 1.2 Related works

**Theoretical understanding why pre-training helps downstream tasks**: Most of the relevant existing works rely on latent variable models and show that pre-training could recover some form of the latent variables. Arora et al. [2016] proposed the RAND-WALK latent model and explains the empirical success of word embedding methods such as word2vec [Mikolov et al., 2013] and GloVe [Pennington et al., 2014]. Arora et al. [2017] extended the previous model to justify sentence embeddings, and Arora et al. [2018] explained sentence embedding via compressed sensing. Other models are also used in this line of work, e.g., hidden Markov models [Wei et al., 2021] and graphical models [Zhang and Hashimoto, 2021]. Lee et al. [2021] and Tosh et al. [2021] assume conditional independence or multi-view structure in the pre-training data and prove that training an additional linear layer on top of learned representations can perform well in downstream tasks.

The problem setting in our paper is similar to that of Saunshi et al. [2021], which also analyzes the performance transfer of pre-trained language models to binary classification downstream tasks. They treat the pre-trained model as a black box and assume that the downstream task can be formulated as a sentence completion task, while we open the black box at the last layer and connect pre-training with downstream tasks by the "anchor vector" and function $f^*$. Moreover, they focus on the prediction probabilities of the pre-trained model while we instead focus on the representations.

**Applications and analysis of hidden representations from large-scale language models**: The hidden representations produced by large language models such as BERT [Devlin et al., 2018] or ELMo [Peters et al., 2018] have been very useful in various NLP tasks. A standard method is to train a linear classifier on these representations, though there are other methods such as using the normalized mean of concatenated word embeddings [Tanaka et al., 2020]. To understand why these word embeddings are useful, people have empirically showed that BERT word embeddings contain information about sentence-level context [Miaschi and Dell'Orletta, 2020], word sense [Wiedemann et al., 2019], and syntactic phenomena [Tenney et al., 2019] including parse trees [Hewitt and Manning, 2019, Kim et al., 2020]. Other empirical explanations include the flatness of local minima achieved by pre-training [Hao et al., 2019], connection to deep metric learning [Tschannen et al., 2019], and the attention patterns across different heads [Kovaleva et al., 2019].

Language modeling can also be considered as a way of using the hidden representations because the next word probability is usually the softmax of the product of the representation and the dictionary matrix. Therefore, any zero-shot application of pre-trained auto-regressive language models, e.g., GPT-3 [Brown et al., 2020] and T5 [Raffel et al., 2020], is a specific method of using the hidden representations.

Some previous works have found that the word embeddings learned by language models can lie in a narrow cone: Gao et al. [2019] empirically found this phenomena for the learned word embeddings in LSTM and vanilla transformers. Ethayarajh [2019], Cai et al. [2020] found similar phenomena for contextualized embeddings in BERT-like models.

## 2 Problem setup

**Notations.** We use $[n]$ to denote the set $\{1, 2, \ldots, n\}$. For an input sequence $x = (x_1, \ldots, x_L)$, we use $x_{-i}$ to denote the input sequence without the $i$-th entry where $i \in [L]$, i.e., $x_{-i} := (x_1, \ldots, x_{i-1}, x_{i+1}, \ldots, x_L)$. We let $D_{KL}(p||q)$ be the KL-divergence between distributions $p$ and $q$ and define $H(p)$ to be the entropy of distribution $p$.

**Ground-truth model.** We consider the following model: There is a set of words $[n]$, each with a fixed corresponding vector $v_j^* \in \mathbb{R}^d$ ($j \in [n]$). We refer to each $v_j^*$ as an atom and the matrix $[v_1^*, \cdots, v_n^*] \in \mathbb{R}^{d \times n}$ as the dictionary. At each position $i$, let $x_i$ be the value of the word at that position; then $x_i \in [n]$. Assume that the probability of $x_i$ given $x_{-i}$ follows a log-linear model, i.e.,

$$p^*(x_i = j | x_{-i}) \propto \exp(\langle v_{-i}^*(x_{-i}), v_j^* \rangle), \tag{1}$$

where $v_{-i}^*(\cdot)$ is a function that encodes the remaining sequence $x_{-i}$ into a vector in $\mathbb{R}^d$.

We also use $z_j^*(x, i) := \langle v_{-i}^*(x_{-i}), v_j^* \rangle$ to denote the $j$-th logit and $Z^*(x, i) := \sum_{j=1}^n \exp(z_j^*(x, i))$ to denote the partition function, i.e., the normalization factor of equation (1). In other words,

$$\forall j \in [n], \quad p^*(x_i = j | x_{-i}) = \frac{\exp(z_j^*(x, i))}{Z^*(x, i)} = \frac{\exp(\langle v_{-i}^*(x_{-i}), v_j^* \rangle)}{Z^*(x, i)}. \tag{2}$$

**Student model.** We use a black-box neural network model whose penultimate layer outputs a $d'$-dimensional vector $v_{-i}(x_{-i}) \in \mathbb{R}^{d'}$ and the last layer is a fully-connected layer with weight matrix $[v_1, v_2, \cdots, v_n] \in \mathbb{R}^{d' \times n}$ followed by the softmax function. In other words, the model output is

$$p(x_i = j | x_{-i}) \propto \exp(\langle v_{-i}(x_{-i}), v_j \rangle). \tag{3}$$

Similar to the ground-truth model, we use $z_j(x, i) := \langle v_{-i}(x_{-i}), v_j \rangle$ to denote the $j$-th logit and $Z(x, i) := \sum_{j=1}^n \exp(z_j(x, i))$ to denote the partition function of equation (3), so

$$\forall j \in [n], \quad p(x_i = j | x_{-i}) = \frac{\exp(\langle v_{-i}(x_{-i}), v_j \rangle)}{Z(x, i)}. \tag{4}$$

**Pre-training.** For self-supervised pre-training, we are given (data, "label") pairs $(x_{-i}, x_i)$, and want our model to predict the "label" $x_i$ given $x_{-i}$. The pre-training loss we use here is cross-entropy loss:

$$\ell(v_{-i}) = \mathbb{E}_x[-p^*(x_i | x_{-i}) \log p(x_i | x_{-i})] = \mathbb{E}_x[D_{KL}(p^*(x_i | x_{-i}) || p(x_i | x_{-i}))] + \mathbb{E}_x[H(p^*(x_i | x_{-i}))].$$

Note that $\mathbb{E}_x[H(p^*(x_i | x_{-i}))]$ is a constant, and we assume that our student model achieves a small loss value so that the KL-divergence term $\mathbb{E}_x[D_{KL}(p^*(x_i | x_{-i}) || p(x_i | x_{-i}))] \leq \epsilon_{\mathrm{KL}}$ for some $\epsilon_{\mathrm{KL}}$.

**Downstream task.** The downstream task we are considering is binary sequence classification, e.g., sentiment classification. For instance, given a sentence, we (probably after adding some prompts) use our pre-trained model to predict the missing word $x_i$ from the given input $x_{-i}$. We assume that there is a perfect classifier $f^*(x, i)$ only depending on $v_{-i}^*$ that can distinguish between positive and negative samples. In other words, for all $x \in \mathrm{POS}$, $f^*(x, i) > 0$ and for all $x \in \mathrm{NEG}$, $f^*(x, i) < 0$. Here POS and NEG are the set of positive and negative input sequences, respectively. We choose to focus on binary classification for theoretical analysis for simplicity, but most of the ideas can be extended to a multi-class classification setting.

A simple downstream task is one whose classifier is linear in $v_{-i}^*$, that is, $f^*(x, i) := \langle v_{-i}^*(x_{-i}), u^* \rangle$ for some $u^* \in \mathbb{R}^d$. One might also expect more structures in the vectors $u^*$ and $v_{-i}^*$. For $u^*$, the downstream task usually depends on only a small set of the words which are related to this task. For example, for sentiment classification, the sentiment of a sentence depends mostly on words similar to "positive" or "negative". This can be formalized as the following assumption:

**Assumption 1** ($u^*$ is a $k$-sparse combination of $\{v_j^*\}_{j=1}^n$). *Assume $u^*$ is a sparse combination of at most $k$ vectors in $\{v_j^*\}_{j=1}^n$ and without loss of generality assume these $k$ vectors are $\{v_1, \ldots, v_k\}$, i.e., there exist coefficients $\{c_j^*\}_{j=1}^k \in \mathbb{R}^k$ such that $u^* = \sum_{j=1}^k c_j^* v_j^*$.*

Note that the fixed representations $\{v_1, \ldots, v_k\}$ correspond to the weights of the last layer in large language models instead of the token representations $v_{-i}^*(x_{-i})$. In other words, the token representations can still be different depending on their context.

There are usually differences between input distributions of pre-training and downstream tasks. The difference may be due to different data sources. For example, the samples in the downstream task

may only contain movie reviews while the pre-training dataset can include all kinds of texts on the Internet. It can also result from prompting, which has become the dominant way of using large language models for downstream tasks [Brown et al., 2020, Radford et al., 2019]. For instance, for movie review classification, appending "This movie was" to the original input could improve the classification accuracy. Similar to the assumption made in Saunshi et al. [2021], we use $\mu \in (0, 1]$ to capture this difference. A smaller $\mu$ indicates a larger difference between the two distributions, and $\mu = 1$ if and only if these two distributions are the same. In most tasks, the two distributions are not too different and we would expect a reasonable value of $\mu$.

**Assumption 2** (Difference between pre-training and downstream distribution). *Let $p_{pre}$ and $p_{DS}$ be the probability density functions of the pre-training and downstream task, respectively. We assume that there exists $\mu \in (0, 1]$ such that*

$$\forall i, \forall x \in \text{POS} \cup \text{NEG}, \; p_{pre}(x_{-i}) \geq \mu \cdot p_{DS}(x_{-i}).$$

If the ground truth classifier $f^*$ can be too close to 0, it would not be robust to small perturbations. We use the standard margin assumption to avoid such cases:

**Assumption 3** (Margin for downstream task). *There exists a margin $\gamma \in \mathbb{R}^+$ such that at any position $i$, if $x \in \text{POS}$, then $f^*(x, i) \geq \gamma$, and if $x \in \text{NEG}$, then $f^*(x, i) \leq -\gamma$, where POS and NEG are the sets of positive and negative samples, respectively.*

Ideally, we want to show that such simple downstream tasks can also be solved well with the representations learned by our student model, i.e., $v_{-i}$ and $\{v_j\}_{j=1}^k$. However, as we will see in Section 3, our current model with this margin assumption still doesn't guarantee good downstream task performances. More structures in the model and the downstream task is necessary to make sure that the pre-trained representations are useful for the downstream task.

# 3 Cases when learned representations are insufficient for downstream tasks

The problem setting in Section 2 seems reasonable at first sight, but in the following subsections, we will show that this model is not enough to guarantee good pre-training performance to generalize to downstream tasks. In other words, there are ways for the student model to approximate the ground-truth probabilities very well in terms of KL divergence but perform very badly at the downstream task. Therefore, we need to put further constraints on the ground-truth model and the downstream task.

## 3.1 Downstream tasks sensitive to words with super-small probability

Intuitively, KL divergence is a weighted log probability difference between two distributions where the weight is the ground-truth probability. Therefore, for the entries with small ground-truth probabilities, a large log probability difference will not result in a large KL divergence. However, the log probability difference is proportional to the difference in the value of $f^*(x, i)$. This makes it possible for the student model to flip the sign of $f^*(x, i)$ without incurring a large KL divergence, as presented in Theorem 1 whose proof is given in Appendix A.

**Theorem 1.** *Suppose the downstream task performance depends only on a function $f^*(x, i) = \langle v_{-i}^*(x_{-i}), u^* \rangle = \sum_{t=1}^k c_t^* \langle v_{-i}^*(x_{-i}), v_t^* \rangle$. For $t^- \in [k]$, define $p^- := p^*(x_i = t^-|x_{-i})$, and assume $p^- \leq \frac{1}{2}$. Then for all $s \in \mathbb{R}^+$, there exist functions $v_{-i}$ and $\{v_t\}_{t=1}^k$ such that $D_{\text{KL}}(p^*(x_i|x_{-i})||p(x_i|x_{-i})) \leq 2sp^-$ and $f(x, i) := \sum_{t=1}^k c_t^* \langle v_{-i}(x_{-i}), v_t \rangle \leq f^*(x, i) - s \cdot c_{t^-}^*$.*

Theorem 1 shows that if there is some word of interest $t^-$ that has a small probability $p^-$, then it is possible to have a model with small KL divergence in pre-training but bad downstream performance. This is because changing the KL divergence by only $2p^- \cdot \frac{f^*(x,i)}{c_{t^-}^*}$ is enough to change the label of the downstream prediction. In other words, as long as the KL divergence is higher than the threshold $2p^- \cdot \frac{f^*(x,i)}{c_{t^-}^*}$, we cannot distinguish between the case where the student model makes an already small probability even smaller (which can hurt the downstream task performance) and the case where random approximation errors are spread across the entries. In this case, a small KL divergence does not necessarily imply good downstream performance.

Note that this sensitivity of the downstream task to very small logits is not natural. For the downstream tasks in practice, after conditioning on the context, whether a word has a probability of $10^{-5}$ or

$10^{-10}$ should not influence the label of the sequence. Thus, we need to impose additional structure on our model. We make the downstream task ignore super-small entries by setting a threshold for the logits and ignoring the logits smaller than that threshold. In this case, making the logits smaller when they are already small will have no influence on the downstream task performance. Concretely, the enhanced model will be ($\sigma(x) := \max\{x, 0\}$ is the ReLU function):

$$f^*(x, i) = \sum_{j=1}^{k} a_j^* \sigma(z_j^*(x, i) - b_j^*) = \sum_{j=1}^{k} a_j^* \sigma(\langle v_{-i}^*(x_{-i}), v_j^* \rangle - b_j^*). \tag{5}$$

### 3.2 Representations are not shift-invariant

The softmax function is invariant under shift, i.e., the output stays the same if we add the same value to every coordinate of the input. In the current model, we have no control over the shift of student model logits on unseen data. Consequently, even if we get a student model that performs well on the training data for the downstream task, we cannot guarantee the performance of this model on new data. This can be formalized in the following theorem.

**Theorem 2.** *Assume $z^*(x, i)$ is bounded. For any function $f^*(x, i) = \sum_{j=1}^{n} a_j \sigma(z_j^*(x, i) - b_j)$, there exist functions $\{\hat{z}_j(x, i)\}_{j=1}^{n}$ such that for all $x$ and $i$, we have $\hat{p}(x_i | x_{-i}) = p^*(x_i | x_{-i})$ and $\hat{f}(x, i) := \sum_{j=1}^{n} a_j^* \sigma(\hat{z}_j(x, i) - b_j^*)$ is always equal to 0. In other words, the pre-training loss of the model $\{\hat{z}_j(x, i)\}_{j=1}^{n}$ is the same as $\{z_j(x, i)\}_{j=1}^{n}$, but its logits are useless for the downstream task.*

*Proof.* We choose $\tau \in \mathbb{R}$ such that $\forall x, i, \tau < \min_{j \in [n]} b_j^* - \max_{j \in [n]} z_j^*(x, i)$, and $\forall x, i, \forall j \in [n]$, we set $\hat{z}_j(x, i) := z_j^*(x, i) + \tau$, then

$$\forall j \in [n], \hat{z}_j(x, i) - b_j^* < z_j^*(x, i) + \min_{j \in [n]} b_j^* - \max_{j \in [n]} z_j^*(x, i) - b_j^* \leq 0, \tag{6}$$

which implies that $\sigma(\hat{z}_j(x, i) - b_j^*) = 0$. Therefore, $\forall x, i$, we have $\hat{f}(x, i) = 0$. □

Theorem 2 indicates that without any structure in the representations, the student model is able to shift the logits for any sample and keep the pre-training loss unchanged. In the worst case, it can shift the logits for unseen data drastically, resulting in a bad downstream performance. Therefore, a theoretical guarantee for downstream performance requires structure in the representations learned by the pre-trained model.

## 4 "Anchor vector" hypothesis and empirical verifications

In Section 3.2, we showed that the shift-invariance of the softmax function can potentially make the student logits useless for the downstream task. Therefore, to understand why the downstream tasks benefit from representations from the pre-trained model, we need to understand the structure of these representations, and this structure must be able to handle the shift-invariance problem.

### 4.1 "Anchor vector" hypothesis

There are different ways to prevent the shift-invariance of softmax from influencing the performance of the downstream tasks. One way of doing this is to keep the partition function stable. Recall that in (4), the probability of a word is the exponential of the corresponding logit divided by the partition function. If the partition function is constant for different samples, the logits can be uniquely determined by the probabilities, which solves the shift-invariance problem. Arora et al. [2016] showed that when both the word embeddings and the latent representation are uniformly distributed on a sphere, the partition function is close to a constant with high probability. They also empirically verified this uniformity of word embeddings trained using GloVe [Pennington et al., 2014] and word2vec [Mikolov et al., 2013] on English Wikipedia. However, as we will see in later experiments in Sections 4.2 and F, this is not true for recent large-scale pre-trained language models.

Instead of uniformity of word embeddings, in large pre-trained models such as GPT-2 [Radford et al., 2019], we observe that if we remove several most frequent words from the computation of the log partition function, the remaining part can be well approximated by the inner product between the

hidden representation $v_{-i}(x)$ and a fixed vector. This motivates us to have the following "anchor vector" hypothesis:

**Definition 1.** *For a sample $x$ and position $i$, we could select a set of bulk words $B(x, i) \subset [n]$, and we define the bulk partition functions as $Z^*_{\text{bulk}}(x, i) := \sum_{j \in B(x,i)} \exp(\langle v^*_{-i}(x_{-i}), v^*_j \rangle)$ and $Z_{\text{bulk}}(x, i) := \sum_{j \in B(x,i)} \exp(\langle v_{-i}(x_{-i}), v_j \rangle)$.*

The selection of bulk words $B(x, i)$ can usually be selected manually or by a simple algorithm. For instance, we can construct $B(x, i)$ by taking out the words corresponding to the largest entries in $p(x_i|x_{-i})$. We can also manually select all the words that are irrelevant to the downstream task.

**Hypothesis 1** ("Anchor vector" hypothesis). *There exists $v_0 \in \mathbb{R}^d$ such that*

$$\langle v_{-i}(x_{-i}), v_0 \rangle \approx \log Z_{\text{bulk}}(x, i).$$

If our hypothesis holds, we can use $v_0$ as an anchor for the logits because it can be used to estimate the partition function, and this does not change the form of the downstream task by much. In the next subsection, we will show that Hypothesis 1 holds for most popular language models and provide some discussions. More details about these models are provided in Appendix E.

## 4.2 Empirical verification of "anchor vector" hypothesis

Figure 1a plots the mean squared approximation error of the log bulk partition function. We use different versions of pre-trained GPT-2 [Radford et al., 2019] and OPT [Zhang et al., 2022], and use the first $1/4$ of WikiText-2 [Merity et al., 2016] as the input text. The hidden representations we use in this experiment are the last hidden states of these models, i.e., the output of the penultimate layer. The dimension of the hidden representations ranges from 768 to 2048, and the number of tokens is about 70k. We choose the bulk words to be all the words except those having top-$k$ probabilities and compute the optimal anchor vector using the closed-form least-squares solution. In our experiments, we use the mean squared error (MSE) to measure the approximation quality. Formally, the MSE is defined as

$$\epsilon_{\text{MSE}} = \min_{v_0} \mathbb{E}_{x,i}[(\langle v_{-i}(x_{-i}), v_0 \rangle - \log Z_{\text{bulk}}(x, i))^2].$$

The values of the bulk partition functions are around 10 (with comparable standard deviation), and we can see from Figure 1a that the MSE is usually several orders of magnitude smaller. Therefore, the inner product between the hidden representation and the optimal anchor vector can usually approximate the log bulk partition function well, and the approximation improves as $k$ increases, i.e., when we ignore more top words. This validates our "anchor vector" hypothesis.

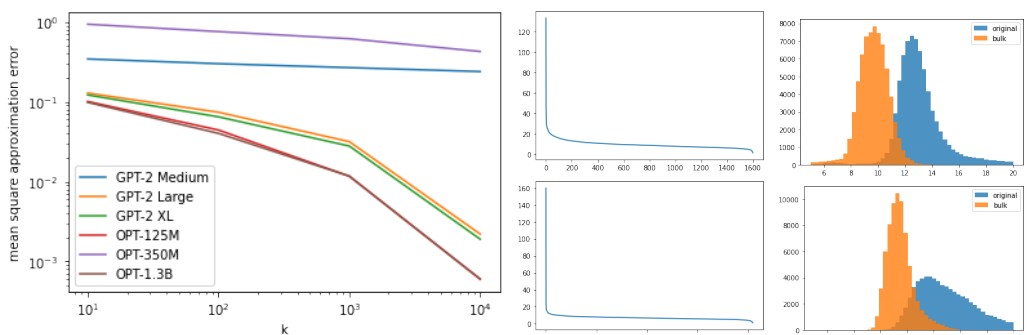

(a) Mean squared error of log bulk partition function with different $k$ in different models. Bulk partition function includes all logits except top-$k$ ones.

(b) Singular values of dictionary matrices (up: GPT-2 XL, down: OPT-1.3B)

(c) Histogram of log original/bulk partition functions (GPT-2 XL/OPT-1.3B)

Figure 1: Empirical verifications of "anchor vector" hypothesis

The existence of anchor vector is quite surprising. Although there are some settings where the partition function can become easy to predict, e.g., when the word embeddings are uniformly distributed on a sphere, we have empirically shown that the large-scale language models considered in this paper

do not fall under these settings. Figure 1b shows that the word embeddings are far from uniformly distributed on a sphere, and Figure 1c indicates that the values of log partition functions and log bulk partition functions vary greatly for different samples. More experiments and discussions are provided in Section F.

# 5 Anchor vector guarantees performance transfer from pre-training to downstream tasks

Based on the counterexamples in Section 3 and the observations in Section 4, we now give sufficient conditions on the downstream tasks so that the downstream classification accuracy provably benefits from pre-trained representations.

## 5.1 Model and assumptions

For huge language models, the anchor vector hypothesis states that there exists a vector $v_0^*$ such that its inner product with the hidden state $v_{-i}^*(x_{-i})$ well approximates the logarithm of the bulk partition function. Therefore, we can use $v_0^*$ as an anchor to handle the shift invariance problem of the softmax function, i.e., we want to subtract $\langle v_{-i}^*(x_{-i}), v_0^* \rangle$ from $\langle v_{-i}^*(x_{-i}), v_j^* \rangle$. As a result, we modify our model for the downstream task to be

$$f^*(x, i) := \sum_{j=1}^{k} a_j^* \sigma(\langle v_{-i}^*(x_{-i}), v_j^* - v_0^* \rangle - b_j^*). \tag{7}$$

The student model is also modified accordingly:

$$f(x, i) := \sum_{j=1}^{k} a_j \sigma(\langle v_{-i}(x), v_j - v_0 \rangle - b_j). \tag{8}$$

Note that in the above model we have already adapted Assumption 1 which was originally stated for the linear model. Therefore, we also update the assumption and restate it below:

**Assumption 4** (At most $k$ words of interest). *Assume there are at most $k$ vectors in $\{v_j^*\}_{j=1}^{n}$ whose logits are relevant to the downstream task and WLOG assume these $k$ vectors are $\{v_1^*, \ldots, v_k^*\}$. In other words, we assume there exist coefficients $\{a_j^*\}_{j=1}^{k} \in \mathbb{R}^k$ such that $f^*(x, i) := \sum_{j=1}^{k} a_j^* \sigma(\langle v_{-i}^*(x_{-i}), v_j^* - v_0^* \rangle - b_j^*)$.*

We are still assuming the teacher-student setting and the log-linear word production model for the word probabilities, as restated below:

$$p^*(x_i = j | x_{-i}) = \frac{\exp(\langle v_{-i}^*(x), v_j^* \rangle)}{Z^*}, \quad p(x_i = j | x_{-i}) = \frac{\exp(\langle v_{-i}(x), v_j \rangle)}{Z}. \tag{9}$$

Under the modified model defined above, we will make some additional assumptions. As noticed in the experiments, the log bulk partition function (as defined in Definition 1) can be linearly approximated by the hidden state. Normally, the bulk only contains words that are not related to the downstream task, and every single word usually has a small probability, but the total probability of the bulk words is not negligible, as reflected in the following two assumptions. Note that the set of bulk words can be strictly contained in the complement of the set of words of interest. This is useful especially when we are not certain about which words are important for the downstream task.

**Assumption 5** (Bulk contains no words of interest). *For all $x$ and $i$, $B(x, i) \cap \{1, \ldots, k\} = \emptyset$.*

**Assumption 6** (Lower bound of bulk probability). *For all $x$ and $i$,*

$$\frac{Z_{\text{bulk}}^*(x, i)}{Z^*(x, i)} \geq p_b.$$

The "anchor" vector $v_0^*(i)$ is used to handle the instability of the partition function, and we formalize our anchor vector hypothesis as the following assumption:

**Assumption 7** (Linear approximation of log bulk partition function). *There exist $v_0, v_0^* \in \mathbb{R}^d, \varepsilon_b \in \mathbb{R}^+$, s.t.,*

$$\forall x, \max\{|\langle v_{-i}^*(x), v_0^* \rangle - \log Z_{\text{bulk}}^*(x, i)|, |\langle v_{-i}(x), v_0 \rangle - \log Z_{\text{bulk}}(x, i)|\} \leq \varepsilon_b.$$

*Furthermore, we assume $\varepsilon_b \leq \frac{\gamma}{4k \max_{j \in [k]} |a_j^*|}$.*

For notational simplicity, we will use all the notations (including $f^*$, $f$, $v_0^*$, and $v_0$) without $i$ when the selection of $i$ is clear from the context.

## 5.2 Main theorem and interpretations

In our model, the ground-truth function $f^*$ contains $k$ terms with coefficients $\{a_j^*\}_{j=1}^k$ and we define the margin $\gamma$ as the margin for $f^*$. If we scale $\{a_j^*\}_{j=1}^k$ or increase $k$ by adding duplicated terms to $f^*$, we can scale $\gamma$ arbitrarily without changing the pre-training performance or the downstream prediction of the student model. To construct a quantity that better indicates the difficulty of the downstream task, we introduce the following definition of the normalized margin that is invariant to the scaling of $k$ and $\{a_j^*\}_{j=1}^k$:

**Definition 2.** *The normalized margin is defined as $\Gamma := \frac{\gamma}{k \max_{j \in [k]} |a_j^*|}$.*

Then we are ready to state our main result. The proof and further discussions for this theorem will be provided in Section B.

**Theorem 3.** *Let $\epsilon_{\text{KL}} := \mathbb{E}_{x \sim \mathcal{D}_{pre}}[D_{\text{KL}}(p^*(x) || p(x))]$ be the pre-training loss, and $\epsilon_{CLS} := \Pr_{x \sim \mathcal{D}_{DS}}[f(x) \cdot f^*(x) < 0]$ be the downstream classification error rate, where $\mathcal{D}_{pre}$ and $\mathcal{D}_{DS}$ are the input distributions of pre-training and downstream data. Under Assumptions 2-7, further assuming $\min_{j \in [k]} b_j^* \geq \epsilon_b - \log(4k)$ and $8\epsilon_b < \Gamma < 6$, there exists a set of parameters $(a_j)_{j=1}^n, (b_j)_{j=1}^n$ such that*

$$\epsilon_{CLS} \leq \epsilon_{\text{KL}} \cdot \frac{288}{\mu \cdot p_b^2 \cdot \Gamma^2}. \tag{10}$$

This theorem shows that when an "anchor vector" exists, we can upper bound the downstream classification error by the KL divergence of the student model during pre-training. This upper bound becomes smaller when the distribution of downstream input is close to that of pre-training, the bulk probability is non-negligible, and the normalized margin of the ground-truth classifier is large. Here we discuss these ways to decrease the upper bound and their corresponding intuitions.

**Large $\mu$.** $\mu$ is larger when the data distributions of pre-training and the downstream task are closer, which helps with the performance transfer.

**Large $p_b$.** The bulk probability $\frac{Z_{\text{bulk}}^*}{Z^*}$ is usually at least a constant in practice. When the bulk probability becomes larger, the anchor vector plays a more important role in the partition function and the downstream task.

**Large $\Gamma$.** A larger normalized margin makes it harder for the student model to make mistakes in downstream prediction.

**Remark 1.** *For ease of presentation, we are making additional assumptions in Theorem 3, e.g., $\min_{j \in [k]} b_j^* \geq \epsilon_b - \log(4k)$ and $8\epsilon_b < \Gamma < 3$. More general cases are covered in Lemma 1. We also discuss potential ways to improve the bound in Section B.1.*

## 6 Conclusions, limitations and future work

In this paper, we analyzed when and how the representations generated by pre-trained large-scale language models can be used in downstream classification tasks. We found two necessary conditions for guaranteeing the usefulness of these representations: the insensitivity of the downstream task to super-small probability words, and the underlying structure in the representations to handle the shift-invariance of softmax. We also provide a sufficient condition, i.e., the existence of an anchor vector, that can guarantee the representations from pre-trained language model to help with downstream tasks. We verify this existence empirically in various large language models and believe that this is an important reason why recent large-scale language models can adapt to different downstream tasks.

We made some assumptions to provide theoretical understanding on how representations can be helpful for downstream applications, and these assumptions may not always hold in practice. For instance, some downstream tasks require multiple rounds of reasoning and do not rely on a small set of words, and the "anchor vector" hypothesis can be weaker for some models such as OPT-350M and GPT-2 Medium. Relaxing these assumptions is a meaningful future direction.

While our work showed the existence of the anchor vector, it remains unclear why this vector exists in most large language models. It might be related to the initialization, optimization, and structure of the neural networks, especially transformers, and it could also be related to the underlying structure of the training data. Digging deeper into this may reveal more fundamental properties of these models. Our analysis of downstream tasks focuses on classification tasks. Since these large networks perform well in various types of downstream tasks, another future direction would be to analyze other kinds of downstream tasks.

We only consider using the last layer representation without fine-tuning the whole model, which is usually weaker in performance. Furthermore, we model the network before the softmax function as a black box, ignoring its inner structure. Further opening up this black box and consider fine-tuning the whole model are important for deeper understanding of the structures in the learned representations from pre-training.

## Acknowledgements

This work is supported by NSF Award DMS-2031849, CCF-1845171 (CAREER) and a Sloan Research Fellowship.

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

# A Proof of Theorem 1

**Theorem 1.** *Suppose the downstream task performance depends only on a function $f^*(x, i) = \langle v^*_{-i}(x_{-i}), u^* \rangle = \sum_{t=1}^{k} c^*_t \langle v^*_{-i}(x_{-i}), v^*_t \rangle$. For $t^- \in [k]$, define $p^- := p^*(x_i = t^- | x_{-i})$, and assume $p^- \leq \frac{1}{2}$. Then for all $s \in \mathbb{R}^+$, there exist functions $v_{-i}$ and $\{v_t\}_{t=1}^{k}$ such that $D_{\mathrm{KL}}(p^*(x_i|x_{-i})||p(x_i|x_{-i})) \leq 2sp^-$ and $f(x, i) := \sum_{t=1}^{k} c^*_t \langle v_{-i}(x_{-i}), v_t \rangle \leq f^*(x, i) - s \cdot c^*_{t^-}.$*

*Proof.* We choose functions $v_{-i}$ and $\{v_{k_t}\}_{t=1}^{r}$ such that $\forall x, i, \forall t \in [n] \setminus \{t^-\}$, $\langle v_{-i}(x_{-i}), v_t \rangle = \langle v^*_{-i}(x_{-i}), v^*_t \rangle$. Besides, $\forall x, i, \langle v_{-i}(x_{-i}), v_{t^-} \rangle = \langle v^*_{-i}(x_{-i}), v^*_{t^-} \rangle - s$.

From the definition of $p^-$ and $t^-$ we know that

$$p^- = p^*(x_i = t^- | x_{-i}) = \frac{\exp(\langle v^*_{-i}, v^*_{t^-} \rangle)}{Z^*(x, i)}. \tag{11}$$

Following our construction of the student model, its partition function satisfies

$$Z(x, i) = \sum_{j=1}^{n} \exp(\langle v_{-i}, v_j \rangle) > \sum_{j \neq t^-} \exp(\langle v^*_{-i}, v^*_j \rangle) \tag{12}$$

$$= Z^*(x, i) - \exp(\langle v^*_{-i}, v^*_{t^-} \rangle) = (1 - p^-)Z^*(x, i). \tag{13}$$

Besides,

$$Z(x, i) = \sum_{j=1}^{n} \exp(\langle v_{-i}, v_j \rangle) < \sum_{j=1}^{n} \exp(\langle v^*_{-i}, v^*_j \rangle) = Z^*(x, i). \tag{14}$$

Therefore,

$$\forall j \neq t^-, \quad \frac{p^*(x_i = j|x_{-i})}{p(x_i = j|x_{-i})} = \frac{\frac{\exp(\langle v^*_{-i}, v^*_j \rangle)}{Z^*(x,i)}}{\frac{\exp(\langle v^*_{-i}, v^*_j \rangle)}{Z(x,i)}} = \frac{Z(x, i)}{Z^*(x, i)}. \tag{15}$$

$$\frac{p^*(x_i = t^- |x_{-i})}{p(x_i = t^- |x_{-i})} = \frac{\frac{\exp(\langle v^*_{-i}, v^*_j \rangle)}{Z^*(x,i)}}{\frac{\exp(\langle v^*_{-i}, v^*_j \rangle - r)}{Z(x,i)}} = \frac{Z(x, i)}{Z^*(x, i)} \cdot e^s. \tag{16}$$

Thus,

$$D_{\mathrm{KL}}(p^*(x_i|x_{-i})||p(x_i|x_{-i})) = \sum_{j=1}^{n} p^*(x_i = j|x_{-i}) \log \frac{p^*(x_i = j|x_{-i})}{p(x_i = j|x_{-i})} \tag{17}$$

$$= \sum_{j \neq t^-} p^*(x_i = j|x_{-i}) \log \frac{Z(x, i)}{Z^*(x, i)} + p^- \cdot s \log \frac{Z(x, i)}{Z^*(x, i)} \tag{18}$$

$$< sp^- \log \frac{1}{1 - p^-} < 2sp^- \tag{19}$$

and

$$f(x, i) = \sum_{t=1}^{k} c^*_t \langle v_{-i}(x_{-i}), v_t \rangle = \sum_{t=1}^{k} c^*_t \langle v^*_{-i}(x_{-i}), v^*_t \rangle - c^*_t \cdot s = f^*(x, i) - s \cdot c^*_{t^-}. \tag{20}$$

$\square$

# B Proof and discussions for main theorem

Here we provide a proof sketch of Theorem 3.

**Theorem 3.** *Let $\epsilon_{\mathrm{KL}} := \mathbb{E}_{x \sim \mathcal{D}_{pre}}[D_{\mathrm{KL}}(p^*(x)||p(x))]$ be the pre-training loss, and $\epsilon_{CLS} := \Pr_{x \sim \mathcal{D}_{DS}}[f(x) \cdot f^*(x) < 0]$ be the downstream classification error rate, where $\mathcal{D}_{pre}$ and $\mathcal{D}_{DS}$*

*are the input distributions of pre-training and downstream data. Under Assumptions 2-7, further assuming* $\min_{j \in [k]} b_j^* \geq \epsilon_b - \log(4k)$ *and* $8\epsilon_b < \Gamma < 6$*, there exists a set of parameters* $(a_j)_{j=1}^n, (b_j)_{j=1}^n$ *such that*

$$\epsilon_{CLS} \leq \epsilon_{\mathrm{KL}} \cdot \frac{288}{\mu \cdot p_b^2 \cdot \Gamma^2}. \tag{10}$$

*Proof.* We have the following lemma providing a lower bound of the TV distance between $p^*(x_i|x_{-i})$ and $p(x_i|x_{-i})$, which will be proved in Section C.

**Lemma 1.** *There exists a choice of* $(a_j)_{j=1}^k, (b_j)_{j=1}^k$ *such that if Assumptions 2-7 hold and there exists* $x \in \mathrm{POS} \cup \mathrm{NEG}$ *and* $i$ *such that* $f(x,i) \cdot f^*(x,i) < 0$*, then we must have*

$$\mathrm{TV}(p^*(x_i|x_{-i}), p(x_i|x_{-i})) \geq p_b \min_{\varepsilon_p \geq 0} \left\{ (1 - e^{-\varepsilon_p}) + k \cdot \min_{j \in [k]} e^{b_j^* - \varepsilon_b - \varepsilon_p} \cdot \left( e^{\sigma\left(\frac{\Gamma}{2} - 2\varepsilon_b - \varepsilon_p\right)} - 1 \right) \right\}. \tag{21}$$

From Lemma 1and Pinsker's inequality we can lower bound the KL divergence at any incorrectly classified sample: For all $x \in \mathrm{POS} \cup \mathrm{NEG}$,

$$\mathrm{KL}(p^*(x_i|x_{-i})||p(x_i|x_{-i})) \tag{22}$$

$$\geq 2(\mathrm{TV}(p^*(x_i|x_{-i}), p(x_i|x_{-i})))^2 \tag{23}$$

$$\geq 2p_b^2 \min_{\varepsilon_p \geq 0} \left\{ (1 - e^{-\varepsilon_p}) + k \cdot \min_{j \in [k]} e^{b_j^* - \varepsilon_b - \varepsilon_p} \cdot \left( e^{\sigma\left(\frac{\Gamma}{2} - 2\varepsilon_b - \varepsilon_p\right)} - 1 \right) \right\}^2 \tag{24}$$

Since the pre-training loss is the expected KL divergence for pre-training data, it is lower bounded by the KL divergence on the incorrectly classified samples, which is related to the downstream classification error rate:

$$\epsilon_{\mathrm{KL}} = \mathbb{E}_{x \sim \mathcal{D}_{pre}}[\mathrm{KL}(p^*(x_i|x_{-i})||p(x_i|x_{-i}))] \tag{25}$$

$$\geq \Pr_{x_{-i} \sim \mathcal{D}_{pre}}[f(x,i) \cdot f^*(x,i) < 0] \cdot \min_{x_{-i}:f(x,i) \cdot f^*(x,i) < 0} \mathrm{KL}(p^*(x_i|x_{-i})||p(x_i|x_{-i})) \tag{26}$$

$$\geq \mu \cdot \Pr_{x_{-i} \sim \mathcal{D}_{DS}}[f(x,i) \cdot f^*(x,i) < 0] \cdot \min_{x_{-i}:f(x,i) \cdot f^*(x,i) < 0} \mathrm{KL}(p^*(x_i|x_{-i})||p(x_i|x_{-i})) \tag{27}$$

$$\geq \epsilon_{CLS} \cdot 2\mu \cdot p_b^2 \min_{\varepsilon_p \geq 0} \left\{ (1 - e^{-\varepsilon_p}) + k \cdot \min_{j \in A(i)} e^{b_j^* - \varepsilon_b - \varepsilon_p} \cdot \left( e^{\sigma\left(\frac{\Gamma}{2} - 2\varepsilon_b - \varepsilon_p\right)} - 1 \right) \right\}^2. \tag{28}$$

Here $\mathcal{D}_{pre}$ and $\mathcal{D}_{DS}$ are the data distributions of pre-training and downstream task, and the second inequality comes from Assumption 2.

Now we have established the relationship between $\epsilon_{\mathrm{KL}}$ and $\epsilon_{CLS}$, and the only step left is to bound the minimum over $\varepsilon_p$. For notation simplicity, we define $\Lambda := \frac{\Gamma}{2} - 2\epsilon_b$.

If $\varepsilon_p \leq \frac{\Lambda}{3}$, we have $(1 - e^{-\varepsilon_p}) \geq (1 - e^{-\frac{\Lambda}{3}})$.

If $\varepsilon_p > \frac{\Lambda}{3}$, we know that

$$e^{b_j^* - \varepsilon_b - \varepsilon_p} \cdot \left( e^{\frac{\gamma}{2k \max_{j \in A(i)} |a_j^*|} - 2\varepsilon_b - \varepsilon_p} - 1 \right) > e^{b_j^* - \varepsilon_b - \frac{\Lambda}{3}} \cdot \left( e^{\frac{2\Lambda}{3}} - 1 \right) = e^{b_j^* - \varepsilon_b} \cdot \sinh\left(\frac{\Lambda}{3}\right).$$

Thus,

$$\min_{\varepsilon_p \geq 0} \left\{ (1 - e^{-\varepsilon_p}) + k \cdot \min_{j \in A(i)} e^{b_j^* - \varepsilon_b - \varepsilon_p} \cdot \left( e^{\sigma\left(\frac{\Gamma}{2} - 2\varepsilon_b - \varepsilon_p\right)} - 1 \right) \right\} \tag{29}$$

$$\geq \min \left\{ 1 - e^{-\frac{\Lambda}{3}}, 2k \cdot \min_{j \in A(i)} e^{b_j^* - \varepsilon_b} \cdot \sinh\left(\frac{\Lambda}{3}\right) \right\} \tag{30}$$

Plugging this into (28) gives us

$$\epsilon_{\mathrm{KL}} \geq \epsilon_{CLS} \cdot 2\mu \cdot p_b^2 \min \left\{ (1 - e^{-\frac{\Lambda}{3}})^2, 4k^2 \cdot \min_{j \in [k]} e^{2b_j^* - 2\varepsilon_b} \cdot \sinh^2\left(\frac{\Lambda}{3}\right) \right\}. \tag{31}$$

Since $\Gamma > 8\epsilon_b$, we know that $\frac{\Gamma}{4} < \Lambda < 3$, so $(1 - e^{-\frac{\Lambda}{3}})^2 > \left(\frac{\Lambda}{6}\right)^2 > \frac{\Gamma^2}{576}$.

Besides, $\min_{j \in [k]} b_j^* \geq \epsilon_b - \log(4k)$ implies $4k^2 \cdot \min_{j \in [k]} e^{2b_j^* - 2\varepsilon_b} \geq \frac{1}{4}$. We also have $\sinh^2(\frac{\Lambda}{3}) > \left(\frac{\Lambda}{3}\right)^2 > \frac{\Gamma^2}{144}$.

Thus, $\min\left\{(1 - e^{-\frac{\Lambda}{3}})^2, 4k^2 \cdot \min_{j \in [k]} e^{2b_j^* - 2\varepsilon_b} \cdot \sinh^2(\frac{\Lambda}{3})\right\} > \frac{\Gamma^2}{576}$. Plugging this into (31) finishes the proof of Theorem 3.

$\square$

## B.1 Further discussions of Theorem 3

**Potentially improving the upper bound by better thresholding.** We are taking the minimum over two terms in the proof of Theorem 3. These two terms come from two parts of the difference between the ground-truth probability and our learned probability: the bulk part and the top part. Both parts are functions of $\epsilon_p$, which measures the approximation error for the bulk probability. A large $\epsilon_p$ will incur a large error in the bulk part while a smaller $\epsilon_p$ will make the top part larger. We are setting $\frac{\Lambda}{3}$ as the threshold for the current bound in Theorem 3. The bound could be improved by choosing other thresholds if we know more about the values of $\{b_j^*\}_{j=1}^k$ and the normalized margin $\Gamma$.

**More general cases for thresholds $\{b_j^*\}_{j=1}^k$ and normalized margin $\Gamma$.** The thresholds $\{b_j^*\}_{j=1}^k$ can be considered as the model's sensitivity to the logits. A smaller threshold indicates a higher sensitivity. In the proof of Theorem 3, we are assuming that the thresholds are not very small. In general cases where the thresholds can be small, an important observation is that: For the same set of samples, decreasing the thresholds by some number $s$ will increase the normalized margin by $s$ as well. Since both the thresholds and the normalized margin are in the exponent in our bound (sinh function is close to exponential when $\Gamma$ is large), these two effects can cancel out to a large degree and don't influence the bound by much.

**Prompt engineering can help with downstream performance.** Prompt engineering can help with the downstream task performance in multiple ways. One direct way is to make the data distribution closer to that of the pre-training stage to increase $\mu$. Furthermore, it can also indirectly help improve the bound by decreasing the number of activated neurons, increasing the margin, etc.

## C Proof of Lemma 1

Here we provide a proof of Lemma 1, with the proofs of the necessary technical lemmas postponed to Appendix D.

**Lemma 1.** *There exists a choice of $(a_j)_{j=1}^k, (b_j)_{j=1}^k$ such that if Assumptions 2-7 hold and there exists $x \in \text{POS} \cup \text{NEG}$ and $i$ such that $f(x, i) \cdot f^*(x, i) < 0$, then we must have*

$$\text{TV}(p^*(x_i | x_{-i}), p(x_i | x_{-i})) \geq p_b \min_{\varepsilon_p \geq 0}\left\{(1 - e^{-\varepsilon_p}) + k \cdot \min_{j \in [k]} e^{b_j^* - \varepsilon_b - \varepsilon_p} \cdot \left(e^{\sigma\left(\frac{\Gamma}{2} - 2\varepsilon_b - \varepsilon_p\right)} - 1\right)\right\}. \tag{21}$$

*Proof.* For all $j \in [n]$, we set $a_j = a_j^*$ and $b_j = b_j^*$. Since $i$ is fixed in the proof, we will omit $i$ when the selection of $i$ is clear from context.

WLOG, assume $x \in \text{POS}$ because otherwise we can flip the sign of all $a_j^*$'s. In this case, from Assumption 3 we know that $f^*(x, i) \geq \gamma$. Therefore, $f(x, i) \cdot f^*(x, i) < 0$ implies $f(x, i) < 0$, so $f^*(x, i) - f(x, i) > \gamma$.

There are two possible ways to make $f(x, i)$ smaller than $f^*(x, i)$: Making the neurons with positive coefficients smaller or making those with negative coefficients larger. The total difference must be at least $\gamma$, so there must exist one sign whose change in value is at least $\frac{\gamma}{2}$. Formally, we decompose the

difference $f^*(x, i) - f(x, i)$ into two terms:

$$f^*(x, i) - f(x, i) = \sum_{j:a_j^* > 0} a_j^* \left( \sigma(\langle v_{-i}^*(x), v_j^* - v_0^* \rangle - b_j^*) - \sigma(\langle v_{-i}(x), v_j - v_0 \rangle - b_j^*) \right) \quad (32)$$

$$+ \sum_{j:a_j^* < 0} |a_j^*| \left( \sigma(\langle v_{-i}(x), v_j - v_0 \rangle - b_j^*) - \sigma(\langle v_{-i}^*(x), v_j^* - v_0^* \rangle - b_j^*) \right). \quad (33)$$

Note that at least one of the two terms on the right-hand side must be at least $\frac{\gamma}{2}$ because their sum is at least $\gamma$. We first consider the first case, i.e., $\sum_{j:a_j^* > 0} a_j^* \left( \sigma(\langle v_{-i}^*(x), v_j^* - v_0^* \rangle - b_j^*) - \sigma(\langle v_{-i}(x), v_j - v_0 \rangle - b_j^*) \right) \geq \frac{\gamma}{2}$. The analysis of the second term is almost the same as the first one.

For notational simplicity, we define

$$\Delta_j := (\langle v_{-i}^*(x), v_j^* - v_0^* \rangle - b_j^*) - \sigma(\langle v_{-i}(x), v_j - v_0 \rangle - b_j^*), \quad (34)$$

(n.b. $\sigma$ only appears in the second term) which implicitly depends on $x$, and and define the set

$$S(x) := \{j : a_j^* > 0, \Delta_j > 0\}. \quad (35)$$

If $\Delta_j > 0$, we must have $\langle v_{-i}^*(x), v_j^* - v_0^* \rangle - b_j^* > 0$ because ReLU is non-negative. This means that the neurons corresponding to the words in $S(x)$ must be activated.

Intuitively, $S(x)$ is the set of words whose corresponding neuron in our student model has a smaller value than the ground-truth model, and $\Delta_j$ for a word $j \in S(x)$ is the difference between these two neurons. In other words, if we want to make a mistake in classifying a sample $x$, $\Delta_j$ is the obstacle for the neuron to overcome, and the sum of all these obstacles must be at least the margin $\gamma$. Formally, we have the following lemma for $S(x)$ and lower bound for $\Delta_j$:

**Lemma 2.** $S(x) \neq \emptyset$.

**Lemma 3.** $\sum_{j \in S(x)} \Delta_j \geq \frac{\gamma}{2 \max_{j \in S(x)} a_j^*}$.

Considering the TV distance, we get

$$\sum_{j=1}^{n} |p^*(x_i = j | x_{-i}) - p(x_i = j | x_{-i})| \quad (36)$$

$$\geq \sum_{j \in B(x, i)} |p^*(x_i = j | x_{-i}) - p(x_i = j | x_{-i})| + \sum_{j \in S(x)} |p^*(x_i = j | x_{-i}) - p(x_i = j | x_{-i})| \quad (37)$$

$$\geq \left| \sum_{j \in B(x, i)} p^*(x_i = j | x_{-i}) - \sum_{j \in B(x, i)} p(x_i = j | x_{-i}) \right| + \sum_{j \in S(x)} |p^*(x_i = j | x_{-i}) - p(x_i = j | x_{-i})| \quad (38)$$

$$= \left| \frac{Z_{\text{bulk}}^*(x, i)}{Z^*(x, i)} - \frac{Z_{\text{bulk}}(x, i)}{Z(x, i)} \right| + \sum_{j \in S(x)} |p^*(x_i = j | x_{-i}) - p(x_i = j | x_{-i})|, \quad (39)$$

where the first inequality comes from Assumption 5, i.e., $S(x) \subseteq [k] \subseteq [n] \setminus B(x, i)$.

We define $\varepsilon_p := \left| \log \frac{Z_{\text{bulk}}^*(x,i)}{Z^*(x,i)} - \log \frac{Z_{\text{bulk}}(x,i)}{Z(x,i)} \right|$, which is the approximation error of the log bulk probability and an important quantity to trade off the two terms in the TV distance. There are two terms in (39). The first term corresponds to the bulk probability and the second term corresponds to the activated neurons. When $\varepsilon_p$ is large, the bulk probability has a large approximation error, resulting in a large TV distance. We lower bound this term in the following lemma:

**Lemma 4.**

$$\left| \frac{Z_{\text{bulk}}^*(x, i)}{Z^*(x, i)} - \frac{Z_{\text{bulk}}(x, i)}{Z(x, i)} \right| \geq p_b (1 - e^{-\varepsilon_p}). \quad (40)$$

The right-hand side of the above lemma is an increasing function of $\varepsilon_p$, and this lower bound becomes large when the approximation error of the bulk probability is large.

In the other regime when $\varepsilon_p$ is small, the bulk partition function must be approximated accurately, i.e., $\frac{Z^*_{\text{bulk}}(x,i)}{Z^*(x,i)} \approx \frac{Z_{\text{bulk}}(x,i)}{Z(x,i)}$. Thus,

$$|p^*(x_i = j|x_{-i}) - p(x_i = j|x_{-i})| = \left| \frac{\exp(\langle v^*_{-i}(x), v^*_j \rangle)}{Z^*(x,i)} - \frac{\exp(\langle v_{-i}(x), v_j \rangle)}{Z(x,i)} \right| \tag{41}$$

$$\approx \frac{Z^*_{\text{bulk}}(x,i)}{Z^*(x,i)} \left| \frac{\exp(\langle v^*_{-i}(x), v^*_j \rangle)}{Z^*_{\text{bulk}}(x,i)} - \frac{\exp(\langle v_{-i}(x), v_j \rangle)}{Z_{\text{bulk}}(x,i)} \right|. \tag{42}$$

From Assumption 7 we know that the bulk partition functions can be accurately approximated using the vectors $v^*_0$ and $v_0$. Thus, $\frac{\exp(\langle v^*_{-i}(x), v^*_j \rangle)}{Z^*_{\text{bulk}}(x,i)} \approx \exp(\langle v^*_{-i}(x), v^*_j - v^*_0 \rangle)$. Therefore, this difference is approximately $|\exp(\langle v^*_{-i}(x), v^*_j - v^*_0 \rangle) - \exp(\langle v_{-i}(x), v_j - v_0 \rangle)| \approx \exp(\langle v_{-i}(x), v_j - v_0 \rangle)(e^{\Delta_j} - 1)$. As $\varepsilon_p$ becomes smaller, the aforementioned approximations become more accurate, making the TV distance suffer from a term that is roughly proportional to $(e^{\Delta_j} - 1)$. This is formalized in the following lemma:

**Lemma 5.** *For all* $j \in S(x)$,

$$|p^*(x_i = j|x_{-i}) - p(x_i = j|x_{-i})| \geq e^{b^*_j - \varepsilon_b - \varepsilon_p} \cdot p_b \cdot \left( e^{\sigma(\Delta_j - 2\varepsilon_b - \varepsilon_p)} - 1 \right). \tag{43}$$

Plugging Lemma 4 and 5 into (39) gives us

$$\text{TV}(p^*(x_i|x_{-i}), p(x_i|x_{-i})) \tag{44}$$

$$\geq \left| \frac{Z^*_{\text{bulk}}(x,i)}{Z^*(x,i)} - \frac{Z_{\text{bulk}}(x,i)}{Z(x,i)} \right| + \sum_{j \in S(x)} \left| \frac{\exp(\langle v^*_{-i}(x), v^*_j \rangle)}{Z^*(x,i)} - \frac{\exp(\langle v_{-i}(x), v_j \rangle)}{Z(x,i)} \right| \tag{45}$$

$$\geq p_b(1 - e^{-\varepsilon_p}) + \sum_{j \in S(x)} e^{b^*_j - \varepsilon_b - \varepsilon_p} \cdot p_b \cdot \left( e^{\sigma(\Delta_j - 2\varepsilon_b - \varepsilon_p)} - 1 \right) \tag{46}$$

$$\geq p_b(1 - e^{-\varepsilon_p}) + p_b \min_{j \in S(x)} e^{b^*_j - \varepsilon_b - \varepsilon_p} \cdot \sum_{j \in S(x)} \left( e^{\sigma(\Delta_j - 2\varepsilon_b - \varepsilon_p)} - 1 \right) \tag{47}$$

$$\geq p_b(1 - e^{-\varepsilon_p}) + p_b \min_{j \in S(x)} e^{b^*_j - \varepsilon_b - \varepsilon_p} \cdot |S(x)| \cdot \left( e^{\sum_{j \in S(x)} \sigma(\Delta_j - 2\varepsilon_b - \varepsilon_p)/|S(x)|} - 1 \right), \tag{48}$$

where the last inequality comes from Jensen's inequality.

Let $g(u,v) := u \cdot (e^{\frac{v}{u}} - 1)$ for $u, v > 0$, then $\frac{\partial g}{\partial u} = e^{\frac{v}{u}} \left( 1 - \frac{v}{u} - e^{-\frac{v}{u}} \right) \leq 0$, so from $|S(x)| \leq k$ we know that

$$|S(x)| \cdot \left( e^{\sum_{j \in S(x)} \sigma(\Delta_j - 2\varepsilon_b - \varepsilon_p)/|S(x)|} - 1 \right) \geq k \cdot \left( e^{\sum_{j \in S(x)} \sigma(\Delta_j - 2\varepsilon_b - \varepsilon_p)/k} - 1 \right).$$

Using the fact that $\sum_i \sigma(y_i) = \sum_i \max\{y_i, 0\} \geq \max\{\sum_i y_i, 0\} = \sigma(\sum_i y_i)$,

$$\sum_{j \in S(x)} \sigma(\Delta_j - 2\varepsilon_b - \varepsilon_p) \geq \sigma\left( \sum_{j \in S(x)} (\Delta_j - 2\varepsilon_b - \varepsilon_p) \right) \geq \sigma\left( \frac{\gamma}{2\max_{t \in S(x)} a^*_t} - 2k \cdot \varepsilon_b - k\varepsilon_p \right), \tag{49}$$

where the last inequality follows from Lemma 3. Therefore,

$$\text{TV}(p^*(x_i|x_{-i}), p(x_i|x_{-i})) \tag{50}$$

$$\geq p_b(1 - e^{-\varepsilon_p}) + p_b \cdot k \min_{j \in S(x)} e^{b^*_j - \varepsilon_b - \varepsilon_p} \cdot \left( e^{\sigma\left( \frac{\gamma}{2k\max_{t \in S(x)} a^*_t} - 2\cdot\varepsilon_b - \varepsilon_p \right)} - 1 \right). \tag{51}$$

Since we do not have any assumption for $\varepsilon_p$, we take the minimum over all possible values of $\varepsilon_p$ and get the following bound:

$$\text{TV}(p^*(x_i|x_{-i}), p(x_i|x_{-i})) \tag{52}$$

$$\geq p_b \min_{\varepsilon_p \geq 0} \left\{ (1 - e^{-\varepsilon_p}) + k \cdot \min_{j \in S(x)} e^{b_j^* - \varepsilon_b - \varepsilon_p} \cdot \left( e^{\sigma\left( \frac{\gamma}{2k \max_{t \in S(x)} a_t^*} - 2 \cdot \varepsilon_b - \varepsilon_p \right)} - 1 \right) \right\}. \tag{53}$$

Above we have derived a lower bound for the TV distance between $p^*$ and $p$ when the first term of (33) is at least $\frac{\gamma}{2}$. When the second term of (33) is at least $\frac{\gamma}{2}$, the proof is symmetric to that of the first term and we only need to exchange the role of parameters from the teacher model and student model and the related definitions. Therefore, with the second term being at least $\frac{\gamma}{2}$, we have

$$\text{TV}(p^*(x_i|x_{-i}), p(x_i|x_{-i})) \tag{54}$$

$$\geq p_b \min_{\varepsilon_p \geq 0} \left\{ (1 - e^{-\varepsilon_p}) + k \cdot \min_{j \in S'(x)} e^{b_j^* - \varepsilon_b - \varepsilon_p} \cdot \left( e^{\sigma\left( \frac{\gamma}{2k \max_{t \in S'(x)} |a_t^*|} - 2 \cdot \varepsilon_b - \varepsilon_p \right)} - 1 \right) \right\}, \tag{55}$$

where $S'(x) := \{ j : a_j^* < 0, \Delta_j' > 0 \}$ and $\Delta_j' := (\langle v_{-i}(x), v_j - v_0 \rangle - b_j^*) - \sigma(\langle v_{-i}^*(x), v_j^* - v_0^* \rangle - b_j^*)$.

Since $S(x) \cup S'(x) \subseteq [k]$, merging (53) and (55) finishes the proof of Lemma 1.

$\square$

# D  Detailed proofs of auxilliary lemmas

**Lemma 2.** $S(x) \neq \emptyset$.

*Proof.* Assume by way of contradiction that $S(x) = \emptyset$. Then by the definition of $S(x)$, we know that for all $j$ such that $a_j^* > 0$,

$$\sigma(\langle v_{-i}^*(x), v_j^* - v_0^* \rangle - b_j^*) - \sigma(\langle v_{-i}(x), v_j - v_0 \rangle - b_j) \tag{56}$$

$$\leq (\langle v_{-i}^*(x), v_j^* - v_0^* \rangle - b_j^*) - \sigma(\langle v_{-i}(x), v_j - v_0 \rangle - b_j) = \Delta_j \leq 0. \tag{57}$$

Therefore,

$$\sum_{j: a_j^* > 0} a_j^* \left( \sigma(\langle v_{-i}^*(x), v_j^* - v_0^* \rangle - b_j^*) - \sigma(\langle v_{-i}(x), v_j - v_0 \rangle - b_j) \right) \leq 0 < \frac{\gamma}{2},$$

which is a contradiction. Thus, $S(x) \neq \emptyset$. $\square$

**Lemma 3.** $\sum_{j \in S(x)} \Delta_j \geq \frac{\gamma}{2 \max_{j \in S(x)} a_j^*}$.

*Proof.* Note that for all $j \in S(x)$, $\langle v_{-i}^*(x), v_j^* - v_0^* \rangle - b_j^* > \sigma(\langle v_{-i}(x), v_j - v_0 \rangle - b_j^*) \geq 0$, so $\sigma(\langle v_{-i}^*(x), v_j^* - v_0^* \rangle - b_j^*) = \langle v_{-i}^*(x), v_j^* - v_0^* \rangle - b_j^*$. Therefore,

$$\sum_{j \in S(x)} a_j^* \Delta_j = \sum_{j \in S(x)} a_j^* \left( (\langle v_{-i}^*(x), v_j^* - v_0^* \rangle - b_j^*) - \sigma((\langle v_{-i}(x), v_j - v_0 \rangle - b_j^*)) \right) \tag{58}$$

$$= \sum_{j \in S(x)} a_j^* \left( \sigma(\langle v_{-i}^*(x), v_j^* - v_0^* \rangle - b_j^*) - \sigma(\langle v_{-i}(x), v_j - v_0 \rangle - b_j^*) \right) \tag{59}$$

$$\geq \frac{\gamma}{2}, \tag{60}$$

which implies

$$\sum_{j \in S(x)} \Delta_j = \frac{\sum_{j \in S(x)} \max_{t \in S(x)} a_t^* \Delta_j}{\max_{t \in S(x)} a_t^*} \geq \frac{\sum_{j \in S(x)} a_j^* \Delta_j}{\max_{t \in S(x)} a_t^*} \geq \frac{\gamma}{2 \max_{t \in S(x)} a_t^*}. \tag{61}$$

$\square$

**Lemma 4.**

$$\left| \frac{Z^*_{\text{bulk}}(x,i)}{Z^*(x,i)} - \frac{Z_{\text{bulk}}(x,i)}{Z(x,i)} \right| \geq p_b(1 - e^{-\varepsilon_p}). \tag{40}$$

*Proof.* If $\varepsilon_p = 0$, both sides of the inequality equal 0, and the inequality holds. When $\varepsilon_p > 0$,

$$\left| \frac{Z^*_{\text{bulk}}(x,i)}{Z^*(x,i)} - \frac{Z_{\text{bulk}}(x,i)}{Z(x,i)} \right| \tag{62}$$

$$= \frac{Z^*_{\text{bulk}}(x,i)}{Z^*(x,i)} \cdot \left| 1 - \frac{Z_{\text{bulk}}(x,i)}{Z(x,i)} \cdot \frac{Z^*(x,i)}{Z^*_{\text{bulk}}(x,i)} \right| \tag{63}$$

$$= \frac{Z^*_{\text{bulk}}(x,i)}{Z^*(x,i)} \cdot \left| 1 - \exp\left( \log \frac{Z_{\text{bulk}}(x,i)}{Z(x,i)} - \log \frac{Z^*_{\text{bulk}}(x,i)}{Z^*(x,i)} \right) \right| \tag{64}$$

$$\geq \frac{Z^*_{\text{bulk}}(x,i)}{Z^*(x,i)} \cdot \left| 1 - \exp\left( -\left| \log \frac{Z_{\text{bulk}}(x,i)}{Z(x,i)} - \log \frac{Z^*_{\text{bulk}}(x,i)}{Z^*(x,i)} \right| \right) \right| \tag{65}$$

$$\geq p_b(1 - e^{-\varepsilon_p}). \tag{66}$$

$\square$

**Lemma 5.** *For all $j \in S(x)$,*

$$|p^*(x_i = j|x_{-i}) - p(x_i = j|x_{-i})| \geq e^{b^*_j - \varepsilon_b - \varepsilon_p} \cdot p_b \cdot \left( e^{\sigma(\Delta_j - 2\varepsilon_b - \varepsilon_p)} - 1 \right). \tag{43}$$

*Proof.* We know that the word probabilities come from a log-linear model, so

$$|p^*(x_i = j|x_{-i}) - p(x_i = j|x_{-i})| = \left| \frac{\exp(\langle v^*_{-i}(x), v^*_j \rangle)}{Z^*(x,i)} - \frac{\exp(\langle v_{-i}(x), v_j \rangle)}{Z(x,i)} \right|.$$

If the probability $p(x_i = j|x_{-i})$ is reasonably large, i.e., its corresponding neuron is activated, then we can use that with $\Delta_j$ to bound this difference. In other words, when $\langle v_{-i}(x), v_j - v_0 \rangle \geq b^*_j$, we know that

$$\left( \langle v^*_{-i}(x), v^*_j \rangle - \log Z^* \right) - \left( \langle v_{-i}(x), v_j \rangle - \log Z \right) \tag{67}$$

$$= \left( \langle v^*_{-i}(x), v^*_j \rangle - \log Z^*_{\text{bulk}} \right) - \left( \langle v_{-i}(x), v_j \rangle - \log Z_{\text{bulk}} + \left( \log \frac{Z^*_{\text{bulk}}}{Z^*} - \log \frac{Z_{\text{bulk}}}{Z} \right) \right) \tag{68}$$

$$\geq \left( \langle v^*_{-i}(x), v^*_j \rangle - \log Z^*_{\text{bulk}} \right) - \left( \langle v_{-i}(x), v_j \rangle - \log Z_{\text{bulk}} \right) - \varepsilon_p \tag{69}$$

$$= \langle v^*_{-i}(x), v^*_j - v^*_0 \rangle - \langle v_{-i}(x), v_j - v_0 \rangle + \left( \langle v^*_{-i}(x), v^*_0 \rangle - \log Z^*_{\text{bulk}} \right) \tag{70}$$

$$\quad - \left( \langle v_{-i}(x), v_0 \rangle - \log Z_{\text{bulk}} \right) - \varepsilon_p \tag{71}$$

$$\geq \Delta_j - 2\varepsilon_b - \varepsilon_p. \tag{72}$$

Note that the last inequality (72) comes from Assumption 7 and the following property of $\Delta_j$:

$$\langle v^*_{-i}(x), v^*_j - v^*_0 \rangle - \langle v_{-i}(x), v_j - v_0 \rangle = \langle v^*_{-i}(x), v^*_j - v^*_0 \rangle - b^*_j - \left( \langle v_{-i}(x), v_j - v_0 \rangle - b^*_j \right) \tag{73}$$

$$\geq \langle v^*_{-i}(x), v^*_j - v^*_0 \rangle - b^*_j - \sigma(\langle v_{-i}(x), v_j - v_0 \rangle - b^*_j) \tag{74}$$

$$= \Delta_j. \tag{75}$$

Then we bound the difference in probabilities: for all $j \in S(x)$,

$$\left| \frac{\exp(\langle v^*_{-i}(x), v^*_j \rangle)}{Z^*} - \frac{\exp(\langle v_{-i}(x), v_j \rangle)}{Z} \right| \tag{76}$$

$$= \frac{\exp(\langle v_{-i}(x), v_j \rangle)}{Z} \cdot \left| \frac{\exp(\langle v^*_{-i}(x), v^*_j \rangle)}{Z^*} \cdot \frac{Z}{\exp(\langle v_{-i}(x), v_j \rangle)} - 1 \right| \tag{77}$$

$$= \frac{\exp(\langle v_{-i}(x), v_j \rangle)}{Z} \cdot \left| \exp\left( \left( \langle v^*_{-i}(x), v^*_j \rangle - \log Z^* \right) - \left( \langle v_{-i}(x), v_j \rangle - \log Z \right) \right) - 1 \right|. \tag{78}$$

Note that

$$\log \frac{\exp(\langle v_{-i}(x), v_j \rangle)}{Z} = \langle v_{-i}(x), v_j \rangle - \log Z \tag{79}$$

$$= \langle v_{-i}(x), v_j - v_0 \rangle + \langle v_{-i}(x), v_0 \rangle - \log Z_{\text{bulk}} + \log \frac{Z_{\text{bulk}}}{Z} \tag{80}$$

$$\geq b_j^* - \varepsilon_b + \log \frac{Z_{\text{bulk}}^*}{Z^*} + \left( \log \frac{Z_{\text{bulk}}}{Z} - \log \frac{Z_{\text{bulk}}^*}{Z^*} \right) \tag{81}$$

$$\geq b_j^* - \varepsilon_b + \log p_b - \varepsilon_p. \tag{82}$$

Moreover, when $\Delta_j - 2\varepsilon_b - \varepsilon_p > 0$, we have by (72) that

$$\left| \exp \left( (\langle v_{-i}^*(x), v_j^* \rangle - \log Z^*) - (\langle v_{-i}(x), v_j \rangle - \log Z) \right) - 1 \right| \tag{83}$$

$$\geq \exp \left( \Delta_j - 2\varepsilon_b - \varepsilon_p \right) - 1. \tag{84}$$

When $\Delta_j - 2\varepsilon_b - \varepsilon_p \leq 0$, we have

$$\left| \exp \left( (\langle v_{-i}^*(x), v_j^* \rangle - \log Z^*) - (\langle v_{-i}(x), v_j \rangle - \log Z) \right) - 1 \right| \geq 0 = e^0 - 1. \tag{85}$$

Thus,

$$\left| \exp \left( (\langle v_{-i}^*(x), v_j^* \rangle - \log Z^*) - (\langle v_{-i}(x), v_j \rangle - \log Z) \right) - 1 \right| \geq e^{\sigma(\Delta_j - 2\varepsilon_b - \varepsilon_p)} - 1. \tag{86}$$

Therefore, when $\langle v_{-i}(x), v_j - v_0 \rangle \geq b_j^*$, we must have

$$|p^*(x_i = j | x_{-i}) - p(x_i = j | x_{-i})| \geq e^{b_j^* - \varepsilon_b - \varepsilon_p} \cdot p_b \cdot \left( e^{\sigma(\Delta_j - 2\varepsilon_b - \varepsilon_p)} - 1 \right). \tag{87}$$

In the second case where $\langle v_{-i}(x), v_j - v_0 \rangle < b_j^*$, we have $\langle v_{-i}(x), v_j \rangle < \langle v_{-i}(x), v_0 \rangle + b_j^*$. The proof for the second case is very similar to that of the first case, with the main difference being we replace $\langle v_{-i}(x), v_j \rangle$ in the first case by $\langle v_{-i}(x), v_0 \rangle + b_j^*$ in the second case.

In the second case, by the definition of $\Delta_j$ we know that $\Delta_j = (\langle v_{-i}^*(x), v_j^* - v_0^* \rangle - b_j^*)$.

Similar to the first case, we can get

$$\left( \langle v_{-i}^*(x), v_j^* \rangle - \log Z^* \right) - \left( \langle v_{-i}(x), v_0 \rangle + b_j^* - \log Z \right) \tag{88}$$

$$= \left( \langle v_{-i}^*(x), v_j^* \rangle - \log Z_{\text{bulk}}^* \right) - \left( \langle v_{-i}(x), v_0 \rangle + b_j^* - \log Z_{\text{bulk}} \right) + \left( \log \frac{Z_{\text{bulk}}^*}{Z^*} - \log \frac{Z_{\text{bulk}}}{Z} \right) \tag{89}$$

$$\geq \left( \langle v_{-i}^*(x), v_j^* \rangle - \log Z_{\text{bulk}}^* \right) - \left( \langle v_{-i}(x), v_0 \rangle + b_j^* - \log Z_{\text{bulk}} \right) - \varepsilon_p \tag{90}$$

$$\geq \langle v_{-i}^*(x), v_j^* - v_0^* \rangle - b_j^* + \left( \langle v_{-i}^*(x), v_0^* \rangle - \log Z_{\text{bulk}}^* \right) - \left( \langle v_{-i}(x), v_0 \rangle - \log Z_{\text{bulk}} \right) - \varepsilon_p \tag{91}$$

$$\geq \Delta_j - 2\varepsilon_b - \varepsilon_p. \tag{92}$$

Therefore, for the difference in probabilities, if $\Delta_j - 2\varepsilon_b - \varepsilon_p > 0$, we have $\left( \langle v_{-i}^*(x), v_j^* \rangle - \log Z^* \right) > \left( \langle v_{-i}(x), v_0 \rangle + b_j^* - \log Z \right)$, so

$$|p^*(x_i = j | x_{-i}) - p(x_i = j | x_{-i})| \tag{93}$$

$$= \left| \frac{\exp(\langle v_{-i}^*(x), v_j^* \rangle)}{Z^*} - \frac{\exp(\langle v_{-i}(x), v_j \rangle)}{Z} \right| \tag{94}$$

$$\geq \frac{\exp(\langle v_{-i}^*(x), v_j^* \rangle)}{Z^*} - \frac{\exp(\langle v_{-i}(x), v_0 \rangle + b_j^*)}{Z} \tag{95}$$

$$= \frac{\exp(\langle v_{-i}(x), v_0 \rangle + b_j^*)}{Z} \cdot \left( \exp \left( (\langle v_{-i}^*(x), v_j^* \rangle - \log Z^*) - (\langle v_{-i}(x), v_0 \rangle + b_j^* - \log Z) \right) - 1 \right) \tag{96}$$

$$\geq \frac{\exp(\langle v_{-i}(x), v_0 \rangle + b_j^*)}{Z} \cdot \left( e^{\Delta_j - 2\varepsilon_b - \varepsilon_p} - 1 \right). \tag{97}$$

If $\Delta_j - 2\varepsilon_b - \varepsilon_p \leq 0$, the absolute value of probability difference can be trivially bounded below by 0. Thus,

$$|p^*(x_i = j | x_{-i}) - p(x_i = j | x_{-i})| \geq \frac{\exp(\langle v_{-i}(x), v_0 \rangle + b_j^*)}{Z} \cdot \left(e^{\sigma(\Delta_j - 2\varepsilon_b - \varepsilon_p)} - 1\right). \quad (98)$$

Since

$$\log \frac{\exp(\langle v_{-i}(x), v_0 \rangle + b_j^*)}{Z} = b_j^* + \langle v_{-i}(x), v_0 \rangle - \log Z_{\text{bulk}} + \log \frac{Z_{\text{bulk}}}{Z} \quad (99)$$

$$\geq b_j^* - \varepsilon_b + \log \frac{Z_{\text{bulk}}^*}{Z^*} + \left(\log \frac{Z_{\text{bulk}}}{Z} - \log \frac{Z_{\text{bulk}}^*}{Z^*}\right) \quad (100)$$

$$\geq b_j^* - \varepsilon_b + \log p_b - \varepsilon_p, \quad (101)$$

we finally get the same bound as the first case:

$$|p^*(x_i = j | x_{-i}) - p(x_i = j | x_{-i})| \geq e^{b_j^* - \varepsilon_b - \varepsilon_p} \cdot p_b \cdot \left(e^{\sigma(\Delta_j - 2\varepsilon_b - \varepsilon_p)} - 1\right). \quad (102)$$

Merging the two cases, we know that for all $j \in S(x)$,

$$|p^*(x_i = j | x_{-i}) - p(x_i = j | x_{-i})| \geq e^{b_j^* - \varepsilon_b - \varepsilon_p} \cdot p_b \cdot \left(e^{\sigma(\Delta_j - 2\varepsilon_b - \varepsilon_p)} - 1\right). \quad (103)$$

This finishes the proof of Lemma 5. $\qquad \square$

# E   Experiment details

**Language models.**   We use various versions of GPT-2 Radford et al. [2019] and OPT Zhang et al. [2022] with number of parameters ranging from 125M to 1.5B. We use the base, medium, large, xl version of GPT-2 which have #parameters 124M / 355M / 774M / 1.5B and hidden dimension 768 / 1024 / 1280 / 1600. For OPT, we use three versions, with #parameters 125M / 350M / 1.3B and hidden dimension 768 / 1024 / 2048. For all these models, the word probability is the softmax of the product of the penultimate layer representation and the dictionary. This is consistent with our theoretical model introduced in Section 2. The parameter settings and performances of these models are shown in Table 1.

Table 1: Parameters and performances of languge models

| Model | #Param | Hidden dim | Perp |
|---|---|---|---|
| GPT-2 | 124M | 768 | 25.92 |
| GPT-2 Medium | 355M | 1024 | 19.19 |
| GPT-2 Large | 774M | 1280 | 17.13 |
| GPT-2 XL | 1.5B | 1600 | 15.34 |
| OPT-125M | 125M | 768 | 25.01 |
| OPT-350M | 350M | 1024 | 19.69 |
| OPT-1.3B | 1.3B | 2048 | 13.16 |

**Dataset.**   We use WikiText-2 Merity et al. [2016] as the text corpus. WikiText-2 has about 280k tokens, and we only use the first 1/4 of it for computational efficiency. The perplexities of the language models on this corpus are shown in Table 1.

# F   Existence of anchor vector is not trivial

In this section, we will empirically verify that the existence of the anchor vector in large-scale language models does not arise from the trivial reasons that we could think of.

**Linear approximation is not accurate for original partition function.**   Table 2 shows the mean squared approximation error of the log bulk partition function for different models and different $k$. For the models whose log bulk partition functions can be well approximated linearly, their log original partition function cannot. This confirms the necessity of removing the words of interest in the partition function.

Table 2: Mean squared approximation error of log bulk partition function for different models and different $k$.

| Model\$k$ | 0 | 10 | 100 | 1000 | 10000 |
|---|---|---|---|---|---|
| GPT-2 | 625.8 | 651.4 | 665.2 | 685.5 | 727.4 |
| GPT-2 M | 0.7647 | 0.3444 | 0.3005 | 0.2683 | 0.2396 |
| GPT-2 L | 0.7296 | 0.1288 | 0.0740 | 0.0317 | 0.0022 |
| GPT-2 XL | 0.7716 | 0.1219 | 0.0648 | 0.0279 | 0.0019 |
| OPT-125M | 0.6688 | 0.1010 | 0.0443 | 0.0117 | 0.0006 |
| OPT-350M | 2.2449 | 0.9393 | 0.7587 | 0.6182 | 0.4275 |
| OPT-1.3B | 0.5234 | 0.0984 | 0.0402 | 0.0117 | 0.0006 |

**Word embeddings are not uniformly distributed on a sphere.** Figure 2 shows the singular values of dictionary matrices from different models. For the four selected models, their word embeddings are close to being low-rank. Therefore, the word embeddings in these models are far from uniformly distributed on a sphere.

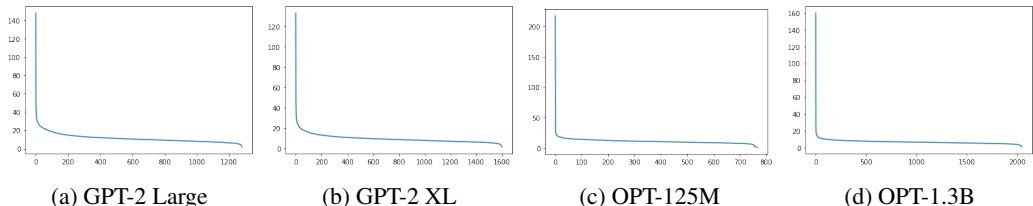

(a) GPT-2 Large     (b) GPT-2 XL     (c) OPT-125M     (d) OPT-1.3B

Figure 2: Singular values of dictionary matrices.

**Dictionary atoms are not uniformly distributed on sphere.** Figure 3 shows the histogram of the $\ell_2$-norm of the dictionary atoms from different language models, and Figure 4 shows the distribution of the Cosine similarity between two random atoms . The norms of these atoms are somewhat bounded, but their cosine similarity is strongly biased towards the positive part, and the dictionary matrices are close to low rank, indicating that these vectors are far from uniformly distributed on a sphere. Instead, they may concentrate around a cone-shaped region.

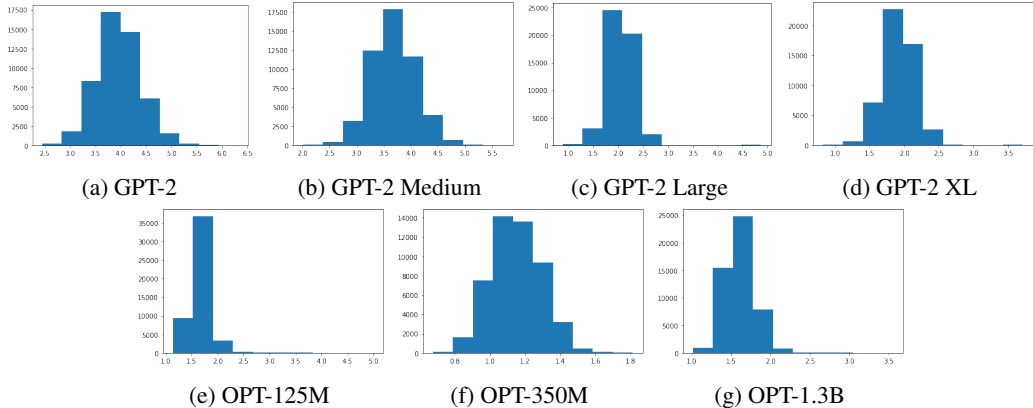

(a) GPT-2     (b) GPT-2 Medium     (c) GPT-2 Large     (d) GPT-2 XL

(e) OPT-125M     (f) OPT-350M     (g) OPT-1.3B

Figure 3: $\ell_2$-norms of atoms

**(Bulk) partition functions have large variations.** Figure 5 shows the histogram of log partition function and log bulk partition function for the language models, where the bulk words are defined as all the words except the top 100, in terms of logit values. We can see from the figures that both the partition function and the bulk partition function vary a lot depending on the samples. Therefore, in recent large-scale language models, the partition functions are not stable across samples.

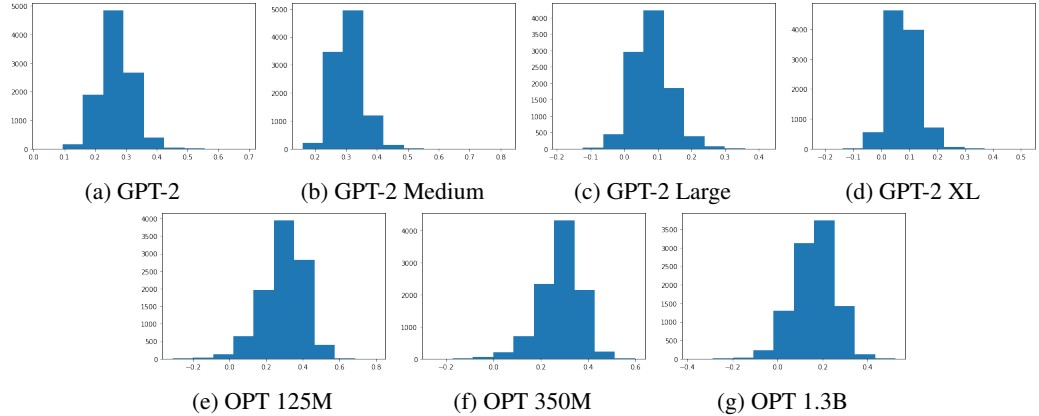

(a) GPT-2    (b) GPT-2 Medium    (c) GPT-2 Large    (d) GPT-2 XL

(e) OPT 125M    (f) OPT 350M    (g) OPT 1.3B

Figure 4: Cosine similarity between two random atoms

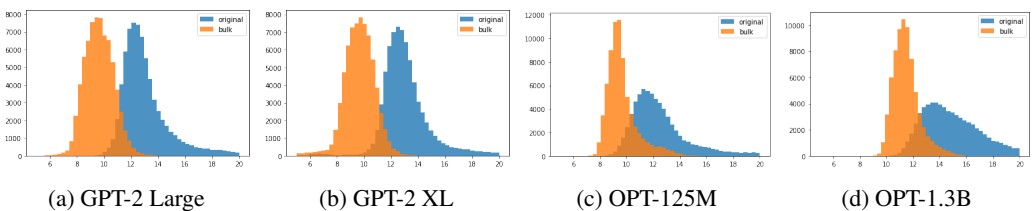

(a) GPT-2 Large    (b) GPT-2 XL    (c) OPT-125M    (d) OPT-1.3B

Figure 5: Histogram of log original and bulk partition functions

