# OpenReview forum: "Connecting Pre-trained Language Model and Downstream Task via Properties of Representation"
_NeurIPS.cc/2023/Conference — NeurIPS 2023 poster_

### Official Review · Reviewer_TyuF · 2023-07-03

**Soundness:** 2 fair
**Presentation:** 3 good
**Contribution:** 2 fair
**Rating:** 6
**Confidence:** 2

**Summary:**

The paper mainly addresses two conditions that enable the representation of pre-trained LLMs to be transferred effectively to the downstream tasks which are usually different from pre-trained objectives.

1. The insensitivity of the downstream task of "super-small" probability words must be guaranteed for good downstream performance. Thus, a function like ReLU is needed as an additional structure of the student model to ignore “small” logits which are considered to have no influence on the downstream task.

2. Using a shift-invariant softmax function can render the logit values meaningless in unseen data. Here, the need for structure in the representation space of the pre-trained model arises and leads to stabilization of the partition function with the “Anchor vector” hypothesis.

The anchor vector hypothesis has backed up with empirical verification. The authors demonstrate that as more words with top probabilities are excluded to compute the optimal anchor vector, the error of approximated bulk partition function decreases. This finding supports that the anchor vector actually exists and can be utilized to address the shift-invariance of the softmax function. Furthermore, with the anchor vector, the upper bound of the downstream task’s error rate can be made.


**Strengths:**

- The paper is well written with a clear flow. Starting from the problem setting, the properties of the learned representation that affect the performance of the downstream task are well stated and addressed with the anchor vector hypothesis.
- The empirical verifications provide substantial evidence in support of the anchor vector hypothesis, solidifying its credibility and reinforcing its validity.


**Weaknesses:**

(1) The empirical verification of the “anchor vector” hypothesis lacks clear explanations and detailed information regarding the experimental setup, leading to confusion and a sense of insufficiency. One particular point that requires clarification is the choice of using auto-regressive models, specifically GPT-2 and OPT, for the verification.

As mentioned earlier in Line 138, the pre-training task of the language model is described as predicting a “label” given a sequence without “label”, which aligns with the mask infilling task associated with masked language models. However, in section 4.2, the verification process utilizes the auto-regressive models, GPT-2 and OPT, which have a pre-training task of predicting the next token of the given sequence.

Based on the provided context, it appears that a single token which can be an intermediate token of the sequence is removed from the sequence (-i) rather than considering the last token of the sequence. This discrepancy raises questions about the alignment between the pretraining task and the verification task depicted in the provided context in section 4.2.
Addressing these issues and providing a clear explanation of the rationale behind using auto-regressive models instead of masked language models for the verification process would enhance the overall understanding of the presented claim.

(2) In Figure 1, the labels for the x and y axes should be provided.


**Questions:**

(1) Based on Figure 1 (a), it is evident that OPT-350M and GPT-2 Medium show minimal decrease in mean squared approximation error, while other three models exhibit a significant decrease in mean squared approximation error as k increases. What could be the underlying reasons for the different tendency? It cannot be attributed solely to the difference in model scale (number of parameters), since OPT-125M displays a considerable decrease in error. Without further analysis, the findings from two of five models (OPT-350M and GPT-2 Medium) present weak evidence to support the “anchor vector” hypothesis. This raises some concerns regarding the generalizability of the proposed hypothesis to pretrained LLMs as a whole.

(2) Do you consider including a broader range of activation functions like swish [1] and gelu [2]? Does the proposed hypothesis still hold for these activation functions, other than ReLU, that do not completely ignore (treat as 0) small logit values?

[1] https://arxiv.org/abs/1710.05941
[2] https://arxiv.org/abs/1606.08415

[1] Ramachandran, P., Zoph, B., & Le, Q. V. (2017). Swish: a self-gated activation function. arXiv preprint arXiv:1710.05941.
[2] Hendrycks, D., & Gimpel, K. (2016). Gaussian Error Linear Units (GELUs). arXiv preprint arXiv:1606.08415.


**Limitations:**

Limitations are listed in section 6: Conclusions and future work.

---

> ### Author Rebuttal · Authors · 2023-08-09
>
> Q: Verification of anchor vector using autoregressive models.
> A: In our definition we allow v*_i to depend on the entire x_{-i} to capture both autoregressive models and masked language models for generality. An autoregressive model can still be a special case of the theory as their v*_i can just depend on the prefix. We chose OPT and GPT-2 mostly because those are the largest models we have access to.
>
> Q: x/y axis of Figure 1
> A: The x axis is the number of highest frequency words that are removed (k in the text), and the y axis is the mean-squared error for the prediction. We will add these labels in the final version.
>
> Q: Figure 1 OPT-350M and GPT-2 Medium.
> A: We honestly don’t know why those models behave that way and we agree with this weakness.
>
> Q: Considering other nonlinearities.
> A: That is a very interesting direction. We suspect it is not difficult to extend our result to swish and GeLU as they are still very flat in the large negative range and that is mostly what we need. However there will certainly be more difficulty in the proof due to the nonlinearity of these functions.

---

> > ### Comment · Reviewer_TyuF · 2023-08-19
> >
> > Thank you for your thoughtful response. The paper presents a compelling theoretical perspective on representation within PLMs, particularly focusing on its mechanism in downstream tasks. While the exploration of the link between LLMs, especially ICL, and the anchor hypothesis is intriguing, it appears that further substantiation is required before its generalization can be confidently supported. To better back up this hypothesis, I suggest conducting additional empirical experiments or in-depth analysis. My assessment remains aligned with the initial score assigned.

---

### Official Review · Reviewer_1JJM · 2023-07-04

**Soundness:** 3 good
**Presentation:** 3 good
**Contribution:** 3 good
**Rating:** 6
**Confidence:** 3

**Summary:**

This paper investigates the relationship between language model pretraining and downstream classification tasks. Under certain assumptions, the author theoretically demonstrated that pre-trained models can guarantee performance on downstream tasks with the existence of proposed “anchor vectors”.

**Strengths:**

1. The research question of “why pre-trained models can help with downstream tasks” is quite important.
2. The paper is generally quite clearly written, the question definition is clear and the mathematical settings are reasonable.
3. For me, the method to connect minimizing the KL divergence during pre-training and the performance of downstream tasks is interesting and valuable.


**Weaknesses:**

1. I think there is a difference between “pretrained embedding” (e.g., word2vec) and “large language models” (e.g., BERT/GPT), also the large language model is not equal to “the last layer embeddings”. In this paper, the author takes the model fixed and takes the extracted representations for analysis, however, fine-tuning the whole model is actually a more common way for models such as BERT/GPT on downstream classification tasks.
2. Some of the claims and assumptions may be incorrect. For example, “We can also manually select all the words that are irrelevant to the downstream task.” in Line 255 and “Assume there are at most $k$ vectors are relevant to the downstream task” in assumption 4. These only hold true for certain tasks like SST-2, which is not generalizable.
3. There exists more works need to be discussed and compared. e.g.,
[1] Visualizing and Understanding the Effectiveness of BERT (EMNLP2019)
[2] Revealing the Dark Secrets of BERT (EMNLP2019)
[3] On Mutual Information Maximization for Representation Learning (ICLR 2020)
4. I see some findings from Appendix F (Existence of anchor vector is not trivial) in the supplementary material, however, most of the conclusions are already found from previous research. e.g.,
[4] How Contextual are Contextualized Word Representations? Comparing the Geometry of BERT, ELMo, and GPT-2 Embeddings (EMNLP 2019)
[5] Representation Degeneration Problem in Training Natural Language Generation Models (ICLR 2019)
[6] Isotropy in the Contextual Embedding Space: Clusters and Manifolds (ICLR 2021)


**Questions:**

See weaknesses.

**Limitations:**

The downstream tasks discussed in this paper are somewhat limited, e.g. the sentiment classification. Some of the assumptions are also based on the importance of words that are related to the task itself, which makes the overall framework not general enough.

---

> ### Author Rebuttal · Authors · 2023-08-09
>
> Q1: Large language model is not just last layer embeddings.
> A: Indeed, this paper only considers the setting where one takes the last layer representation of a language model and uses that in downstream tasks without fine tuning the whole model. While this gives reasonable performance for many tasks, it is indeed often weaker than fine tuning the whole model (although it’s significantly cheaper). We will highlight this limitation in the final version.
>
> Q2: Some assumptions may only hold for certain tasks and not generalize.
> A: In general, we agree that we make many assumptions and they may not always hold in practice. The goal of our paper is to give some theoretical understanding on how representations can be helpful for downstream applications, and unfortunately this is extremely difficult without assumptions. We tried to give explanations/motivations for every assumption we make, but we agree that they certainly don’t cover all the settings. We will add more discussions on the limitations of these assumptions.
>
> We also thank the reviewer for the pointers to more references in weaknesses 3 and 4, we will add discussions to these very relevant works.

---

> > ### Comment · Reviewer_1JJM · 2023-08-11
> >
> > I have read the rebuttal and the comments by other reviewers. As the authors have replied, the limitations and relationship between existing work should be added. At present, I maintain my rating unchanged.

---

### Official Review · Reviewer_joK7 · 2023-07-07

**Soundness:** 3 good
**Presentation:** 3 good
**Contribution:** 3 good
**Rating:** 6
**Confidence:** 3

**Summary:**


This paper presents a sequence of assumptions along with their respective conclusions, which advance the objective of comprehending the intricate connection between the performance of pre-training and downstream tasks in language models.
By framing the prevailing language models as log-linear models, this paper initially presents a mathematical demonstration showcasing instances where pre-trained models encounter limitations in effectively transferring their knowledge to address downstream tasks.
Additionally, the paper enumerates several imperative prerequisites for successful knowledge transfer, such as the requirement for shift invariance.
Furthermore, the authors introduce the "anchor vector hypothesis," which serves as a crucial framework for elucidating the remarkable adaptability of language models to various tasks.

**Strengths:**

- Attempted to rigorously prove the effectiveness of pre-training language models with a mathematical framework.
- The claims posited in the paper appear to be reasonable, albeit necessitating a potential round of further verification, provided that readers accept the underlying assumptions upon which the proofs are based.
- The storyline of the paper is intuitive and persuasive.





**Weaknesses:**

- Given the non-negligible series of assumptions outlined in the paper, a legitimate concern arises regarding the practical implications associated with the claims put forth.
For instance, the authors commence the proof procedures by assuming that "a downstream task depends on only a small set of words related to this task," which may not hold true for tasks that involve complex rounds of reasoning.
- In addition to Section 4.2, it would be advantageous for the paper to include additional empirical experiments that substantiate the claims presented, thereby strengthening the overall argument.


**Questions:**

If we examine the standard (masked) language modeling during the pre-training phase, as far as my understanding goes, $p^*(x_i|x_{-i})$ should correspond to an instance of the Dirac delta distribution. If this understanding is correct, I'm wondering the subsequent discussions in the paper, such as the one in Section 3.1 where scenarios are considered where $p^*$ is relatively small but possibly not zero, are still valid.


**Limitations:**

While the mathematical framework proposed by the authors is compelling, there is a valid concern regarding the explanatory power of the paper's contents in relation to the inner workings of current language models. This concern arises due to the series of (unrealistic) assumptions upon which the framework is built.

---

> ### Author Rebuttal · Authors · 2023-08-09
>
> Q: Many assumptions
> A: In general we agree that we make many assumptions and they may not always hold in practice. The goal of our paper is to give some theoretical understanding on how representations can be helpful for downstream applications, and unfortunately this is extremely difficult without assumptions. We tried to give explanations/motivations for every assumption we make, but it is of course still subjective on whether those are sufficient or not. We will add more discussions on the limitations of these assumptions.
>
> Q: p*(x_i|x_{-i}) being a dirac delta
> A: Note that x_i and x_{-i} are discrete (they are words) instead of vector representations. If they are vector representations (which we denote as v*_i(x_{-i}) as in Equation (1)) then that is indeed often deterministic/dirac delta distributions. The distribution p*(x_i|x_{-i}) is the distribution of a word given its context. It usually is not a dirac delta function as that would mean we are absolutely certain which word needs to be at position i given its context, which is rarely true given the inherent amount of ambiguity in language. None of the language models we used in our experiments give dirac delta distributions for p*(x_i|x_{-i}). As for the specific instance of Section 3.1, Theorem 1 is talking about a word that has low probability, which will exist even if p*(x_i|x_{-i}) is almost concentrated on a single word.

---

### Official Review · Reviewer_8QyB · 2023-07-18

**Soundness:** 3 good
**Presentation:** 3 good
**Contribution:** 3 good
**Rating:** 6
**Confidence:** 4

**Summary:**

This paper explores how to connect pretraining performance with downstream task performance (i.e., binary classification). The theoretical analysis is based on token representations. The authors find the ``anchor vector'' in the representation space and bridge pretraining and downstream tasks performance based on it.

---

Rebuttal response:

Thanks for the clarification! It would be better if these parts can be described clearly in the paper.

As for the responses to the two questions, I don't think they are convincing. As I mentioned, previous work used ``surprisal'' to measure the information amount brought by that word. “low probability words” could be vital and carry essential information in that sentence. But this is a theoretical analysis paper. So, I'm fine with this assumption. Therefore, I would only slightly raise my score.

**Strengths:**

This paper discusses a vital problem in language model understanding from a theoretical view. The idea of the proofs is interesting.

**Weaknesses:**

1. There are many assumptions and some of them may not hold in practice. A1 could fail for contextualized language models (e.g., BERT, GPT-2) since one token could have very different representations with different contexts. A4 may fail since the prediction head of downstream tasks could be nonlinear. Besides, there are some claims that could be wrong (see Q1, Q2).


2. The ``anchor vector'' hypothesis could fail. Even if the authors show that the MSE for the approximation is less than 1. However, it is not sufficient to prove the hypothesis empirically. Language models could have quite similar representations for all tokens, and they may be far from the zero point. More baselines should be included to support the hypothesis. For example, calculating the average/minimum MSE distance between two random words.

3. The scope of this paper is not a little limited. Even if in the introduction and empirical section (4.2) popular language models (e.g., GPT-2) are included, the theoretical discussion only considers binary classification as the downstream task. Besides, not sure if I understand it correctly, the analysis of language models assumes that they are uncontextualized (e.g., GloVe and ELMo).

**Questions:**

Q1 (Line 204-206): Why does this claim hold? First of all, the vocabulary size of LM could be very large (e.g., for BLOOM, its vocabulary size is 250,000). A word with probability 1e-5 is not super-small. Second, words with small probability could be vital to the downstream tasks. Some people use ``surprisal'' to measure the information gained from that word. It means words with small probability could convey information.

Q2 (Line 246-248): Does it mean that the frequent words are not that useful? Is the claim contradictory to the aforementioned one?

**Limitations:**

Yes

---

> ### Author Rebuttal · Authors · 2023-08-09
>
> We thank the reviewer for the review. However, there is likely some serious misunderstanding which we try to clarify below:
>
> Q: Many assumptions; some of them can fail.
> A: In general we agree that we make many assumptions and they may not always hold in practice. The goal of our paper is to give some theoretical understanding on how representations can be helpful for downstream applications, and unfortunately this is extremely difficult without assumptions. We tried to give explanations/motivations for every assumption we make, but it is of course still subjective on whether those are sufficient or not.
>
> As for the concrete assumptions, we want to emphasize that A1 *does not* need to fail for contextualized language models. It is indeed true that for these models, the representation of a token can be different depending on their context. However, the “vectors” in A1 refer to the weights of the last layer for these large language models, which are fixed. More precisely, referring to Equation (1) in the paper, the representation of a token is the v*_{-i}(x_{-i}) which is indeed allowed to depend on the context (x_{-i}), while the vectors v*_j’s are the weights of the last softmax layer and are therefore fixed. We believe this is a major confusion that the reviewer had and we will clarify this in the final version.
>
> Q: Anchor vector assumption might fail.
> A: Although we provided empirical evidence that anchor vector assumption holds for the language models we considered, it is indeed possible that for different language models it can fail. However, we would like to point out that the possibility mentioned in the review (that anchor vector assumption might be true because the language model has similar representations for all tokens) is ruled out by our experiments: as we can see from Figure 1, when we choose k to be small (i.e., we do not exclude the frequent words) the approximation guarantee is much worse. This is not possible if all the words have similar representations and is a stronger indicator than measuring the MSE between random words.
>
> Q: Limited scope: binary classification, simple language models.
> A: As the analysis is already quite complicated for binary classification, we choose to focus on that for simplicity. However, most of the ideas can be extended to a multi-class classification setting. The paper does not just apply to simple language models and we suspect the misunderstanding is similar to the one we explained in the first question - we will make sure to clarify this point carefully.
>
> Q1: Why does 204-206 hold?
> A: We think there is again a misunderstanding here. When we talk about “low probability words”, we are not talking about the probability of the word in the dataset/without context. Of course, as the reviewer pointed out, a word with probability 10^-4 is still fairly frequent when we don’t condition on context, and it’s unreasonable to leave those words out. However, in this discussion by “low probability words” we mean the word has low probability after *conditioning” on the context. For example, when we use a prompt like “this movie is ***” for a sentiment classification task, we expect the blank to be some of the words that are extremely relevant to the task (such as good, bad, exciting, boring). For a word that is meaningless in this context (say “for”), we shouldn’t care whether the model predicts it with low or extremely low probability. We believe this is justified in practice because the distribution of words changes significantly after conditioning, and in applying language models people frequently use top-k words after conditioning for k that is not large. We will make this more clear.
>
> Q2: Line 246-248, does that mean that the high probability words are not useful?
> A: No, in fact it means quite the opposite - the high probability words have such a strong influence on the partition function, so that if we don’t remove them, we cannot well-approximate the log-partition function.

---

### Decision · Program_Chairs · 2023-09-21

**Decision:**

Accept (poster)

**Comment:**

The paper explores the connection between downstream task performance and pretraining performance. The authors show the existence of an anchor vector that serves as a critical element to ensure the performance on downstream tasks.